# *Toxoplasma gondii* actin filaments are tuned for rapid disassembly and turnover

Kelli L. Hvorecny[1,4], Thomas E. Sladewski[2,4], Enrique M. De La Cruz [3], Justin M. Kollman [1] ✉ & Aoife T. Heaslip [2] ✉

The cytoskeletal protein actin plays a critical role in the pathogenicity of the intracellular parasite, *Toxoplasma gondii*, mediating invasion and egress, cargo transport, and organelle inheritance. Advances in live cell imaging have revealed extensive filamentous actin networks in the Apicomplexan parasite, but there are conflicting data regarding the biochemical and biophysical properties of *Toxoplasma* actin. Here, we imaged the in vitro assembly of individual *Toxoplasma* actin filaments in real time, showing that native, unstabilized filaments grow tens of microns in length. Unlike skeletal muscle actin, *Toxoplasma* filaments intrinsically undergo rapid treadmilling due to a high critical concentration, fast monomer dissociation, and rapid nucleotide exchange. Cryo-EM structures of jasplakinolide-stabilized and native (i.e. unstabilized) filaments show an architecture like skeletal actin, with differences in assembly contacts in the D-loop that explain the dynamic nature of the filament, likely a conserved feature of Apicomplexan actin. This work demonstrates that evolutionary changes at assembly interfaces can tune the dynamic properties of actin filaments without disrupting their conserved structure.

The human pathogen *Toxoplasma gondii* (*T. gondii*) causes life-threatening disease in immunocompromised individuals and when infection occurs *in utero*[1–3]. *T. gondii* is a member of the Apicomplexan phylum that encompasses over 5000 species of parasites including a large number of parasites of medical and veterinary importance, including *Cryptosporidium spp.* and *Plasmodium spp.*, the causative agents of cryptosporidiosis and malaria respectively.

*T. gondii* is an obligate intracellular parasite. Survival and disease pathogenesis are dependent on host cell invasion, intracellular replication, and egress, which destroys the infected cells. *T. gondii* expresses a single divergent actin gene (*Tg*Act1) that is essential for the successful completion of this lytic cycle. This protein shares only 83% identity to skeletal alpha (α) actin and mammalian beta (β) and gamma (γ) isoforms[4] and was originally shown to be essential for the gliding motility of the parasite, host cell invasion, and egress[5–8]. More recent studies have greatly expanded our understanding of other essential functions of actin in intracellular parasites. These include: the inheritance of a non-photosynthetic plastid organelle named the apicoplast[9,10], the directed movement of secretory vesicles called dense granules[11], recycling of secretory vesicles called micronemes in dividing parasites[12], morphology and positioning of Golgi and post-Golgi compartments, the movement of ER tubules[13] and conoid extension[14,15].

Despite its ubiquitous functions in Apicomplexan parasites, the organization of F-actin in *T. gondii* remained elusive for many years because conventional actin probes such as phalloidin, GFP-tagged actin, and LifeAct failed to detect filamentous structures[16]. The absence of filamentous actin was further supported by biochemical studies which showed that recombinantly purified HIS-*Tg*Act1 formed only short unstable filaments in vitro (reviewed in ref. 16). This is also the case for the closely related *Plasmodium falciparum* actin 1, which shares 93% sequence identity with *Tg*Act1[4].

[1]Department of Biochemistry, University of Washington, Seattle, WA, USA. [2]Department of Department of Molecular and Cell Biology, University of Connecticut, Storrs, CT, USA. [3]Department of Molecular Biophysics and Biochemistry, Yale University, New Haven, CT, USA. [4]These authors contributed equally: Kelli L. Hvorecny, Thomas E. Sladewski. ✉e-mail: jkoll@uw.edu; aoife.heaslip@uconn.edu

The conclusion that Apicomplexan actin does not polymerize into long filaments in vivo has recently been reconsidered given the advances made with the development of an actin chromobody as a tool for imaging F-actin in *T. gondii* and *P. falciparum*[17–19]. Using this approach, studies have revealed vast tubular actin networks that connect individual parasites within host cells and remarkably dynamic actin networks within the parasite cytosol[18,19]. The success of using the actin chromobody is likely because it shows minimal effects on actin dynamics compared to other probes[20,21]. Recently, recombinantly purified actin chromobody has been used as a tool to study the dynamics of recombinantly purified *P. falciparum* actin in vitro and showed that it can polymerize into long dynamic filaments[22]. These results are more consistent with the observation of filamentous networks in the cell and indicate that assembly factors are not required for its polymerization.

While these studies have provided novel insights into *Tg*Act1 organization, many biochemical and structural questions remain. There is conflicting data on whether *Tg*Act1 has a high or low critical concentration, as both have been reported for Apicomplexan actin[22–24]. It has also been reported that *Tg*Act1 filaments assemble isodesmically, and therefore lack a critical concentration[23]. Intriguingly, long filaments of *Tg*Act1 or *P. falciparum* Act1 have rarely been observed directly using electron microscopy without filament-stabilizing agents such as jasplakinolide, which has led to the idea that long filaments of *Tg*Act1 only form in the presence of such stabilizing agents[25]. A lack of high-resolution structures of unstabilized *Tg*Act1 or *P. falciparum* Act1 filaments has also hindered the exploration of the unusual properties of Apicomplexan actin.

In this work, we used epifluorescence microscopy to image the growth of *Tg*Act1 filaments in vitro. We find that *Tg*Act1 can form filaments tens of microns in length that rapidly treadmill. This feature is due to an unusually rapid subunit dissociation constant at the pointed-end and a nucleotide exchange rate constant from the monomer that is 50-fold faster than that of skeletal muscle actin. Using electron cryomicroscopy, we determined the structures of both jasplakinolide-stabilized and native, unstabilized *Tg*Act1 filaments. In these two conditions, we found that the DNase I-binding loop (D-loop) adopts different conformations, with the unstabilized conformation likely contributing to rapid filament subunit dissociation. Taken together, this study not only resolves questions in the field regarding the ability of *Tg*Act1 to form long filaments and its assembly properties, but also demonstrates how dynamic properties can evolve within the constraints of a filament architecture.

## Results

### TgAct1 polymerizes into long filaments in vitro with a high critical concentration

Because modifications to actin, such as tags or direct labeling, can influence the biochemical and kinetic properties of actin polymerization, we purified unmodified *Tg*Act1. Untagged *Tg*Act1 (ToxoDB: TgME49_209030) was prepared by expressing the gene in *Sf*9 cells fused to a C-terminal β-thymosin–6xHIS tag, which maintains the actin in a monomeric state during expression and allows for removal by chymotrypsin, resulting in an untagged and unmodified protein product (Fig. 1a, b and Supplementary Fig. 1a). This strategy was previously used for the expression of *P. falciparum* and *Dictyostelium* actin, which shares 95 and 85% identity to *Tg*Act1 respectively[22,26,27].

Polymerization of individual actin filaments was imaged in real-time using an epifluorescence microscopy-based in vitro actin polymerization assay (Fig. 1c). The fluorophore, a recombinantly purified actin chromobody tagged with EmeraldFP (Supplementary Fig. 1b, hereafter referred to as chromobodies) was previously used to observe filament dynamics and organization in *T. gondii* cells[17], and as a novel method for imaging in vitro polymerization of *P. falciparum* actin[22]. The advantage of this approach is that actin polymerization can be visualized without fluorescent labeling of actin monomers, which has been shown to dramatically affect skeletal actin assembly and disassembly rate constants[28]. Consistent with this behavior, we find that *Tg*Act1 conjugated directly with rhodamine to Cys374 shows poor incorporation into filaments (Supplementary Fig. 1c), and is thus unsuitable for visualizing *Tg*Act1 polymerization in vitro.

Using the chromobody approach, we observe robust *Tg*Act1 polymerization at concentrations above 12 μM actin, with filaments growing to greater than 60 μm in length (Fig. 1d–f and Supplementary Movie S1). To demonstrate that *Tg*Act1 can form long filaments in the absence of the chromobody, we polymerized filaments in a flow chamber without chromobodies for 10 min. Actin monomers were then washed out to stop assembly and chromobodies were added and imaged before filaments disassembled. When we do this, we observe filaments longer than 10 μm, indicating that *Tg*Act1 can polymerize into long filaments in the absence of chromobodies or other stabilizing agents such as jasplakinolide (Supplementary Fig. 2).

Strikingly, over the concentration range measured (12–24 μM), individual filaments treadmill, depolymerizing at the pointed-end while elongating at the barbed-end (Fig. 1e and Supplementary Movie S2). Treadmilling is readily visualized in kymographs (distance vs time), which shows pointed-end disassembly (left) and barbed-end elongation (right) of representative filaments at an actin concentration of 16 μM (Fig. 1g). This property is also a characteristic of *P. falciparum* actin[22] and is thus a likely conserved feature of Apicomplexan actins, which are all divergent from skeletal muscle actin[4,29]. Long-term imaging of actin polymerization shows that filaments continue to polymerize at a constant rate for 20 min of imaging (Fig. 1f).

The actin concentration-dependence of assembly and disassembly rates provides the critical concentration ($C_c$) and the rate constants for subunit incorporation and dissociation at the barbed and pointed ends of filaments. A linear fit to plots of the assembly or disassembly versus actin concentration can be used to determine the critical concentration (x-intercept or ratio of rate constants ($k_-/k_+$)), subunit dissociation rate constant (y-intercept) and subunit association rate constant (slope) at each end of the filament[30]. As a control, we imaged the polymerization of different concentrations of skeletal muscle actin using the chromobody and found that the critical concentration (0.09 μM), disassembly rate constant (1.2 s$^{-1}$) and assembly rate constant (12.7 μM$^{-1}$·s$^{-1}$) are consistent with previously reported values[30], which is further evidence that the chromobodies do not influence the polymerization, critical concentration, or subunit incorporation rate constants (Fig. 1h, orange and Table 1).

To verify that the chromobody has no effects on the rates of *Tg*Act1 assembly, we measured length distributions for *Tg*Act1 assembled in the presence or absence of actin chromobody at different time intervals for three different concentrations of actin using an endpoint actin assembly assay (Supplementary Fig. 2). From this analysis, we find no detectable differences in length distributions with or without chromobody for all time intervals and actin concentrations tested, consistent with the chromobody having no effect on the rate of actin growth (Supplementary Fig. 3a, b and Supplementary Table 1). To quantify these results, we determined an average actin assembly growth rate for each actin concentration with or without chromobody. There was no significant difference in assembly rates due to the chromobody at all actin concentrations tested (Supplementary Fig. 3c).

The elongation rate constants and critical concentration of *Tg*Act1 deviate significantly from those of skeletal muscle actin. For the *Tg*Act1 barbed-end, the critical concentration is 6.5 ± 1.2 μM (Fig. 1h, blue and Table 1) which is >60-fold higher than that of skeletal muscle actin under similar conditions[30]. Determining the $C_c$ by a ratio of rate constants ($k_-/k_+$) was consistent with values calculated from the x-intercept for both skeletal and *Tg*Act1 demonstrating internal consistency (Table 1). We find that the subunit dissociation rate constant at

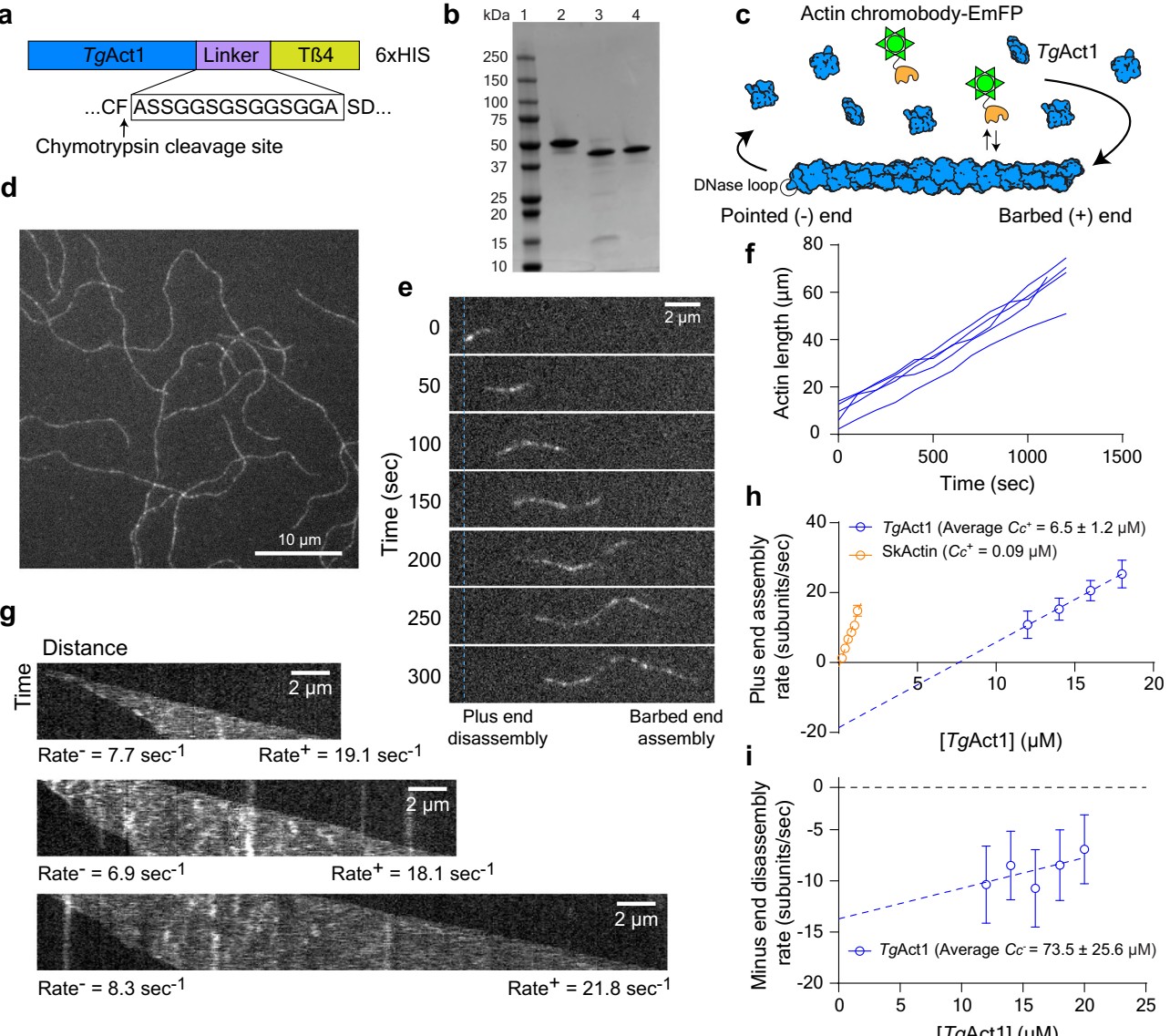

**Fig. 1 | Imaging and analysis of *Tg*Act1 polymerization in vitro. a** Domain organization of *Tg*Act1 fused to a β-thymosin–HIS tag and purification strategy of untagged *Tg*Act1 from *Sf*9 cells. **b** Coomassie-stained SDS-PAGE gel of (lane 1) protein molecular weight marker; (lane 2) *Tg*Act1–β-thymosin–HIS following HIS purification; (lane 3) *Tg*Act1 after chymotryptic digest; (lane 4) purified *Tg*Act1 after ion-exchange and size exclusion chromatography (*N* = 3). **c** Schematic of the in vitro polymerization assay. *Tg*Act1 monomers (blue) were induced for polymerization and added to a blocked flow chamber absorbed with NEM-myosin II. Epifluorescence microscopy was used to image the *Tg*Act1 filament dynamics by inclusion of 25–50 nM actin chromobody fused to EmeraldFP (EmFP) (green). **d** Epifluorescence microscopy image of *Tg*Act1 filaments in the in vitro polymerization assay showing filament lengths >60 µm (*N* = 3). **e** Montage of a single *Tg*Act1 filament shrinking from the pointed (−) end and growing from the barbed (+) end in the presence of 16 µM *Tg*Act1 monomers. The Blue dashed line indicates the position of the (−) end at time zero (*N* = 3). **f** Measuring the filament length of five representative filaments over time shows a constant growth rate over 1200 s. **g** Kymographs for three individual filaments in the presence of 16 µM *Tg*Act1 monomers showing disassembly from the pointed (−) end and assembly from the barbed (+) end. Rates for individual filaments are shown in subunits per second (*N* = 3). **h** Plot of the rate of barbed (+) end growth in subunits/sec, per actin concentration for *Tg*Act1 (blue) compared to skeletal actin (orange). The data shown is an aggregate of three independent preps presented as mean values ± SD. The average critical concentration ($C_c$), determined by the x-intercept of the fitted line, is higher for *Tg*Act1 (6.5 µM) compared to skeletal actin (0.09 µM). **i** Plot of barbed (−) end disassembly rate per *Tg*Act1 concentration. The average $C_c$ for the barbed-end, determined by the x-intercept of the fitted line, is 73.5 µM. Error for critical concentrations is the standard deviation of the mean for three independent *Tg*Act1 preparations. Source data are provided as a Source Data file.

the barbed-end of TgAct1 filaments is $15.1 \pm 2.1\,s^{-1}$ and the elongation rate constant ($k_+$) is $2.3 \pm 0.3\,\mu M^{-1} \cdot s^{-1}$ indicating that the barbed-end of *Tg*Act1 disassembles ~17 times faster and assembles approximately three times slower compared to skeletal muscle actin[30], thereby accounting for the higher critical concentration.

A linear fit of the pointed-end disassembly rate versus *Tg*Act1 concentration shows that the observed critical concentration (x-intercept, $73.5 \pm 25.6\,\mu M$) is ~60-fold higher than the pointed-end of skeletal muscle actin, which arises from a subunit dissociation rate

constant (y-intercept, $10.6 \pm 1.4\,s^{-1}$) that is 33-fold faster than skeletal actin[31] and a slow assembly rate constant ($0.2 \pm 0.04\,\mu M^{-1}\,s^{-1}$) (Fig. 1i, blue and Table 1).

Given the high critical concentration, we measured the cellular concentrations of *Tg*Act1 to assess if they are sufficient to support polymerization in vivo, in the absence of polymerization factors. As shown by western blot (Fig. 2a), T. *gondii* has a relatively high cytosolic actin concentration compared to other cell types (Fig. 2b), and this concentration would be sufficient to drive actin polymerization from

**Table 1 | Summary of polymerization rate constants for *Tg*Act1 and skeletal actin**

| | Critical concentration- tion C$_c$ (μM) x-intercept; k$_-$/k$_+$ | Disassembly k$_-$ (subunits·sec$^{-1}$) | Assembly k$_+$ (subunits·μM$^{-1}$·s$^{-1}$) | n |
|---|---|---|---|---|
| ***Tg*Act1 barbed (+) end rate constants** | | | | |
| Prep 1 | 6.4; 6.4 | 13.4 | 2.1 | 228 |
| Prep 2 | 7.8; 7.9 | 17.4 | 2.2 | 373 |
| Prep 3 | 5.4; 5.4 | 14.6 | 2.7 | 51 |
| Average | 6.5 ± 1.2; 6.6 ± 1.3 | 15.1 ± 2.1 | 2.3 ± 0.3 | |
| **Skeletal actin barbed (+) end rate constants** | | | | |
| | 0.09; 0.09 | 1.2 | 12.7 | 278 |
| ***Tg*Act1 pointed (−) end rate constants** | | | | |
| Prep 1 | 84.3; 108 | 10.8 | 0.1 | 111 |
| Prep 2 | 44.3; 45.5 | 9.1 | 0.2 | 95 |
| Prep 3 | 91.9; 119 | 11.9 | 0.1 | 43 |
| Average | 73.5 ± 25.6; 90.8 ± 40 | 10.6 ± 1.4 | 0.2 ± 0.04 | |

Rate constants for the barbed (+) and pointed (−) end of *Tg*Act1 and skeletal actin were determined from plots of assembly (barbed-end) or disassembly (pointed-end) rates versus actin concentration. The critical concentration (C$_c$) is defined by either the x-intercept or ratio of rate constants (k$_-$/k$_+$). The assembly (k$_+$) and disassembly (k$_-$) rate constants are determined from the slope and y-intercept respectively. Error ± SD. Polymerization conditions: 25 mM imidazole, pH 7.4, 50 mM KCl, 2.5 mM MgCl$_2$, 1 mM EGTA, 2.5 mM MgATP, 10 mM DTT, 0.25% methylcellulose, 2.5 mg/mL BSA, 0.5% Pluronic F127, oxygen scavenging system (0.13 mg/mL glucose oxidase, 50 μg/mL catalase, and 3 mg/mL glucose), 37 °C. Source data are provided as a Source Data file.

both filament ends because the cellular concentration is above the critical concentration for polymerization at both filament ends.

## Rapid nucleotide turnover and exchange allows for intrinsic TgAct1 filament treadmilling

Next, we evaluated how the ATPase properties of *Tg*Act1 differ from skeletal actin to understand how filament treadmilling is maintained. We polymerized *Tg*Act1 and monitored inorganic phosphate (Pi) release using an MESG assay[32,33]. For skeletal actin, the amount of phosphate released plateaus at an equimolar amount of F-actin (Fig. 2c, orange). This was expected because once F-actin formation plateaus, the rate of filament subunit disassembly, nucleotide exchange, and reincorporation is negligible on this timescale. That is, polymerized subunits undergo a single ATPase turnover. Since nucleotide exchange does not occur in monomers incorporated into a filament, polymerized subunits generate inorganic phosphate once, and then must disassemble from the filament for nucleotide exchange to occur[34–36].

Strikingly, time courses of *Tg*Act1 phosphate release do not plateau, and each *Tg*Act1 releases (on average) 3 phosphates over the 1-hour time course (Fig. 2c, blue). This indicates that each actin subunit hydrolyzes multiple ATP molecules during the timescale of the experiment, consistent with *Tg*Act1 subunit treadmilling. As a control, we measured the Pi release rate over a range of actin concentrations (Fig. 2d). As expected, the Pi release rate increases with actin concentration.

MgADP-actin subunits that dissociate from filaments exchange bound MgADP for MgATP before reincorporating to the barbed-end[31,34,36,37]. For skeletal actin, the rate constant for nucleotide exchange is slow[38–42] and generally requires exchange factors, such as profilin, for recharging the monomers with MgATP on a rapid timescale[43–45]. Because *Tg*Act1 rapidly treadmills in the absence of exchange factors (Figs. 1e, 2c), we hypothesized that nucleotide exchange from *Tg*Act1 must be faster compared to skeletal actin. To test this hypothesis, ATP was exchanged for ethenoATP (MgεATP) which displays a fluorescence enhancement when bound to actin. The

dissociation rate constant of εATP was determined by competing bound MgεATP a large molar excess of unlabeled MgATP and monitoring the change in fluorescence over time[41,42,46]. The εATP dissociation from skeletal actin was slow ($k_{-εATP} = 0.003 ± 0.0002$ s$^{-1}$) (Fig. 2e, f, orange) consistent with previous results[35,40,47]. In contrast, the exchange of εATP from *Tg*Act1 was over 50 times faster ($k_{-εATP} = 0.16 ± 0.03$ s$^{-1}$) (Fig. 2e, f, blue). The fast nucleotide exchange explains how *Tg*Act1 can rapidly exchange bound nucleotides and become competent for filament assembly, even without nucleotide exchange factors. Attempts to measure the dissociation of MgεADP were unsuccessful because we did not observe enhanced fluorescence of MgεADP following nucleotide exchange. We reasoned that either the affinity of MgεADP to *Tg*Act1 was too weak or *Tg*Act1 bound to MgεADP is unstable[48].

## Optimization of experimental conditions for cryo-electron microscopy of Toxoplasma actin filaments

Next, we sought to use cryo-electron microscopy to determine the structural basis for the distinct polymerization properties of *Tg*Act1. Initial negative stain micrographs in 0.1 mM ATP incubated for 1 hour at 37 °C with or without jasplakinolide reproduced what has been reported in the literature[23,49]; long filaments are visible with jasplakinolide, whereas short filaments assemble without jasplakinolide (Supplementary Fig. 5a, b). Substituting the stabilizing, non-hydrolysable analog AMPPNP for ATP in the polymerization buffer produced filaments without jasplakinolide (Supplementary Fig. 5c), but we were hesitant to use a non-native ligand for structural exploration because inclusion of 0.1 mM AMPPNP in the actin growth assay resulted in an ~1.5-fold increase in the disassembly rate from the barbed-end and 2-fold increase in the critical concentration compared to MgATP conditions (Supplementary Fig. 4).

Based on our kinetic data, we tested filament conditions that might slow filament disassembly. Shortening the incubation period and lowering the assembly temperature to 25 °C did not produce any filaments (Supplementary Fig. 5d). Decreasing the incubation time and increasing the ATP concentration produced promising results. At time points between 10 and 30 min, long filaments are observed in negative stain, with some filaments extending several microns (Fig. 3a, b). This observation suggests that due to the treadmilling, ATP hydrolysis and nucleotide exchange are rapid enough for *Tg*Act1 to consume the majority of the 0.1 mM ATP in 1 h in the initial reaction conditions. Consistent with this, phosphate release upon *Tg*Act1 polymerization does not plateau at any given *Tg*Act1 concentration (Fig. 2d), and given the observed phosphate release rates (Fig. 2d), 33 μM *Tg*Act1 should hydrolyze 0.1 mM MgATP in 1 h. Decreasing the incubation time and increasing the ATP concentration allowed us to circumvent this outcome. Thus, to capture long filaments in cryo conditions, we froze *Tg*Act1 samples after a 10 to 20-min incubation period with 1 mM ATP (Fig. 4a).

## Structure of unstabilized ADP-bound TgAct1 filaments reveal altered ᴅ-loop positioning and decreased buried surface area

We determined the structure of unstabilized *Tg*Act1 filaments at 2.5 Å resolution, which revealed an overall architecture similar to skeletal and other actin filament structures (Fig. 4b, Supplementary Figs. 6, 7A, and Table 2). *Tg*Act1 and skeletal muscle actin share 83% sequence identity[4], and the structures show many similarities. As we identified MgADP in the nucleotide-binding site (Supplementary Fig. 7b), we compared *Tg*Act1 to structures of rabbit and chicken skeletal muscle actin bound to MgADP (PDB IDs 8d13 and 8a2t)[50,51]. A single *Tg*Act1 protomer is in nearly the same conformation as skeletal actin, with Cα RMSD values of 0.5 and 0.8 for chicken and rabbit actin, respectively (Supplemental Movie 3). The helical symmetry of TgAct1, 28.1 Å rise and −167.1° twist, is nearly identical to chicken and rabbit MgADP-actin

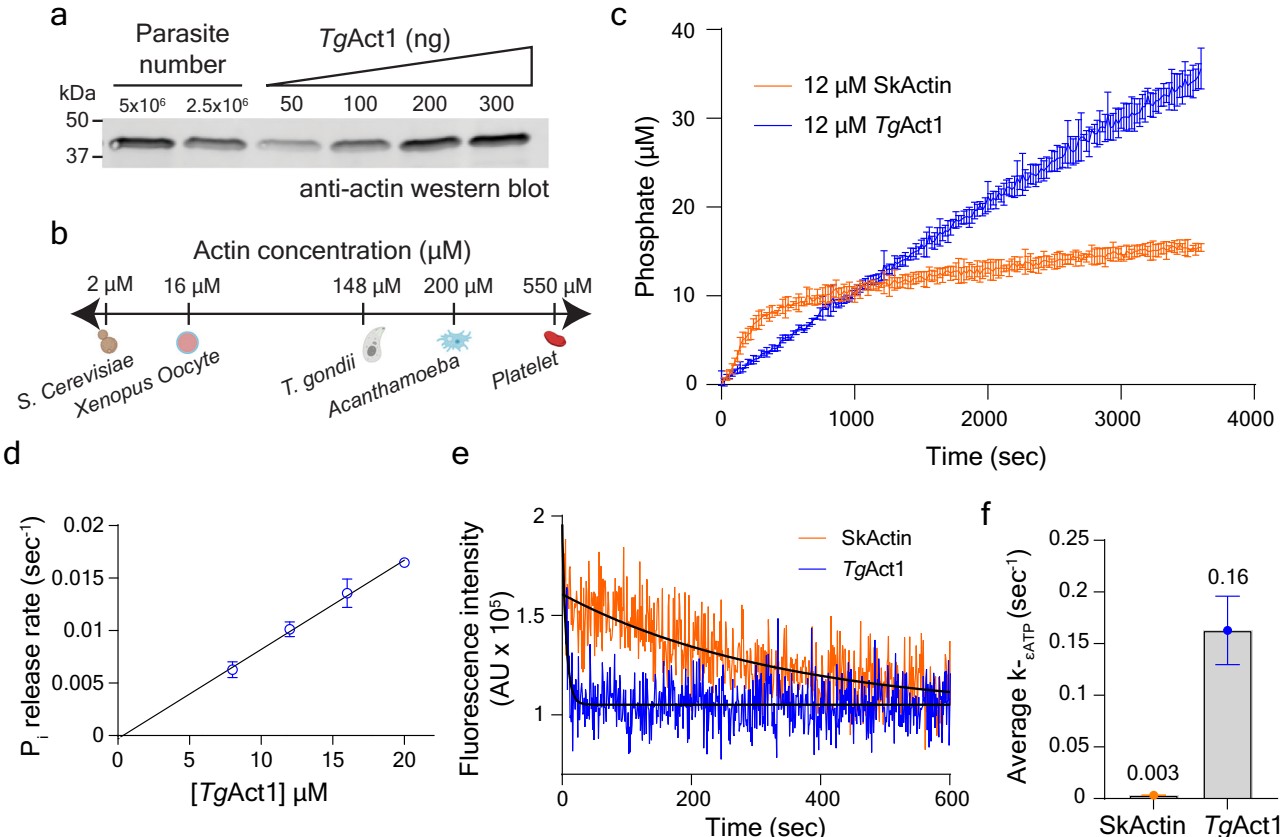

**Fig. 2 | Cellular concentration and kinetic analysis of *Tg*Act1. a** Anti-actin western blot of *T. gondii* parasite lysate and known amounts of purified *Tg*Act1. The average cellular concentration of *Tg*Act1 (148 ± 7.6 μM) was determined by comparing the relative band intensity of a known number of parasites to those of known amounts of purified *Tg*Act1. Error ± SEM, $n = 4$ independent experiments from two independent lysates. **b** Comparison of *Tg*Act1 cellular concentrations to other known model systems[84]. Cartoons created with BioRender.com. **c** Time course of phosphate release from 12 μM *Tg*Act1 (blue) and skeletal actin (orange) after inducing polymerization. Error bars represent the standard deviation of the mean for three independent experiments. **d** Plot of $P_i$-release rates over a range of *Tg*Act1 concentrations. Conditions: 25 mM Imidazole, pH 7.4, 50 mM KCl, 1 mM EGTA, 2 mM MgCl₂, 0.2 mM MgATP, 1 mM DTT, 37 °C. Data were presented as mean values ± SD, $n = 3$ independent experiments. **e** Time courses of fluorescence change after mixing a large molar excess of MgATP with an equilibrated mixture of actin monomers with bound MgεATP. The solid black lines through the data represent the best fits single exponentials, yielding the rate constant for εATP dissociation ($k_{-εATP}$) from *Tg*Act1 (blue) and skeletal actin (orange). Conditions: 20 μM actin, 25 mM Imidazole, pH 7.4, 50 mM KCl, 1 mM EGTA, 2 mM MgCl₂, 1 mM DTT, 0.5 mM MgATP, 37 °C. **f** Bar graph of average $k_{-ATP}$ for TgAct1 (blue, 0.16 ± 0.03) and skeletal actin (orange, 0.003 ± 0.0002). Error ± SD, $n = 3$. Source data are provided as a Source Data file.

filaments, at 28.1 and 27.5 Å rise, respectively, and −166.7° twist for both.

However, there are notable differences between the structures. The most striking difference is in the position of the D-loop (residues 40–51 in *Tg*Act1 and residues 39–50 in skeletal muscle actin), which forms important longitudinal contacts between actin protomers (Supplementary Fig. 7c, d). In *Tg*Act1 filaments, the D-loop has shifted, moving the loop partially out of the pocket where the loop is typically bound, with a Cα RMSD value of 2.1 when compared to chicken skeletal actin[50]. There are six amino acid substitutions on the D-loop in *Tg*Act1 relative to skeletal actin (Supplementary Fig. 8). Two of these substitutions, amino acids Pro42 and Ile44, point in the opposite direction relative to the comparable residues in skeletal actin (Gln41 and Val43) (Fig. 4c). While Val43 of skeletal actin sits within a hydrophobic portion of the binding pocket, the comparable residue in *Tg*Act1 (Ile44) is displaced by 5 Å and does not make inter-protomer contacts (Fig. 4d, e). These changes in the D-loop conformation break most of the backbone interactions that occur among skeletal actin D-loop residues 41–44 (*Tg*Act1 42–44), and hydrogen bond contacts between side chains are lost due to a combination of residue substitution and mainchain displacement (*Tg*Act1 amino acids 39–41 and 46) (Fig. 4f, g and Supplementary

Fig. 9a, b). In contrast to the substitutions on the D-loop, the amino acids in the pocket of the longitudinally adjacent protomer where the D-loop binds are identical between *Tg*Act1 and skeletal actin (Fig. 4f, g). Overall, this reduces the buried surface area in this interaction by about one-third, from 638 Å² in chicken skeletal actin (PDB ID 8d13) to 485 Å² in *Tg*Act1. The D-loop position and resulting decrease in longitudinal protomer contacts likely contribute to the decreased filament stability of *Tg*Act1, explaining the rapid depolymerization rate constant and high critical concentration.

Other structural differences between *Tg*Act1 and skeletal actin filaments are more modest. The C-terminal phenylalanine of TgAct1 occupies a different position than skeletal actin and in *Tg*Act1 it does not interact with the D-loop as it does in skeletal actin (Fig. 4f, g). The nucleotide sensing residue Gln138 (skeletal actin Gln137) and the residues that transmit nucleotide state to the periphery (Glu108 and Arg117; skeletal actin Glu107/Arg116) are conserved in *Tg*Act1. While the residues show positions that don't fully align with either chicken and rabbit MgADP-actin filaments (PDB IDs 8d13 and 8a2t), they may still function to transmit nucleotide state information to the periphery[51] (Supplementary Fig. 9c). Tyr54 from TgAct1 (skeletal actin Tyr53) establishes a hydrogen bond with Tyr170 (skeletal actin Tyr169) on the neighboring protomer (Supplementary Fig. 9a, b). There are

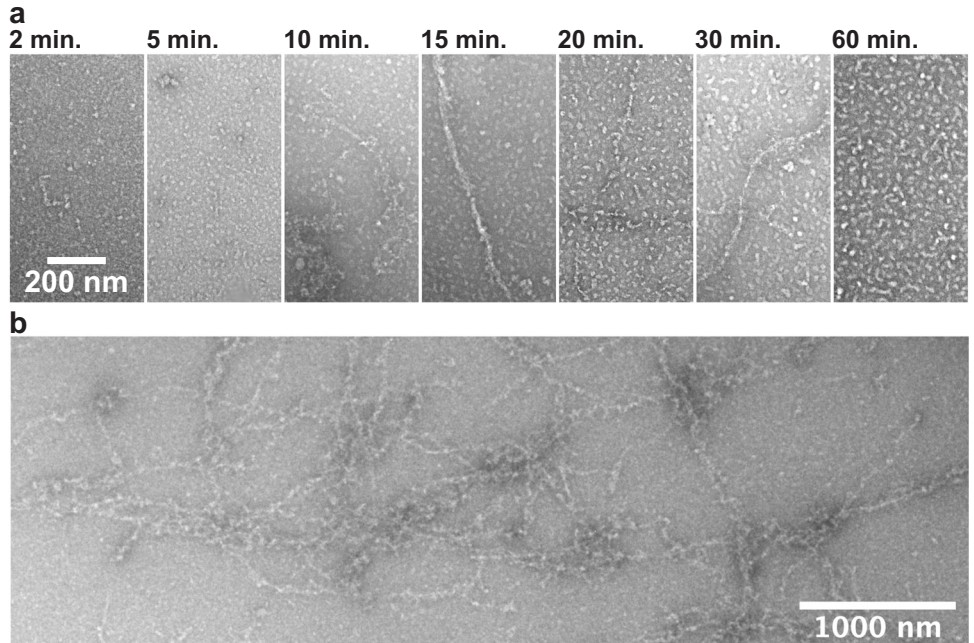

**Fig. 3 | Negative stain electron microscopy of *Tg*Act1 filaments. a** *Tg*Act1 filaments were observed in conditions where the concentration of MgATP was increased from 0.1 to 1 mM and shorter incubation time points were checked (10–30 min). **b** Increased numbers and lengths of filaments were identified on grids assembled at the 20-min time point where the water washes in between protein application and addition of uranyl formate were skipped. For each condition, one grid was prepared and a minimum of two separate locations on each grid were imaged.

four substitutions on the hydrophobic loop (*Tg*Act1 residues 263–275), but the mainchain retains the same conformation. The substituted side chains only make intraprotomer contacts; however, one of the substitutions (*Tg*Act1 Lys270−skeletal actin Met 269) adds a hydrogen bond with the backbone of Asp180, subtly shifting the mainchain around residues 177-180 (Supplementary Fig. 9d). *Tg*Act1 filaments do not contain an extensive water network within the center of the filament (Supplementary Fig. 9e). Comparing the total buried surface area, excluding waters, for one protomer of *Tg*Act1 versus one protomer of skeletal actin provides a summation of the changes: *Tg*Act1 loses 364 Å$^2$ of buried surface area relative to skeletal actin filaments (8d13: 3115 Å$^2$; TgAct1: 2751 Å$^2$), the majority of which is from the change in the position of the D-loop (Fig. 4h, i).

### The D-loop of Jasplakinolide-stabilized TgAct1 filaments resembles the D-loops of unstabilized skeletal and stabilized *P. falciparum* actins

*Tg*Act1 has 93% sequence identity to *Plasmodium falciparum* actin 1, and the amino acids from the D-loops and the pocket where the D-loops bind are identical (Supplementary Fig. 8). Despite this identity, the D-loops of native (i.e., unstabilized) *Tg*Act1 and *P. falciparum* Act1 stabilized with jasplakinolide have different conformations (Fig. 5a and Supplementary Fig. 9f). The D-loop conformation of jasplakinolide-stabilized *P. falciparum* Act1 is similar to the D-loops of both unstabilized skeletal actin filaments and the second *P. falciparum* actin isoform, Act2. Skeletal actin has a distinct D-loop conformation when bound to jasplakinolide (Supplementary Fig. 9g). This suggests that the effect of jasplakinolide on actin filaments are not uniform that jasplakinolide effects on *Tg*Act1 may differ from skeletal or *P. falciparum* actins.

To test this hypothesis, we solved a structure of *Tg*Act1 bound to jasplakinolide at 3.0 Å (Fig. 5b, Supplementary Figs. 5e, 7a–f, and Table 2). Jasplakinolide-bound filaments retained the same helical parameters as unstabilized filaments (28.1 Å rise and −167° twist). The nucleotide-binding site contained MgADP (Supplementary Fig. 7b) and the protomers of the filaments are very similar, with a Cα RMSD of

0.5 Å. As expected, jasplakinolide binds in the same location in *Tg*Act1 filaments as it is found in skeletal and *P. falciparum* filaments (Supplementary Fig. 7f). It primarily bridges the lateral interaction between protomers and increases the lateral surface contacts from 702 Å$^2$ in unstabilized filaments to 1016 Å$^2$ in jasplakinolide-bound filaments.

The D-loop conformation of jasplakinolide-stabilized filaments is distinct from the unstabilized *Tg*Act1 filaments (Fig. 5c, d, Supplementary Fig. 7c, d, and Supplementary Movie 4). With jasplakinolide bound, amino acids Pro42 and Ile44 point into the binding pocket, though the volume does suggest some mixed occupancy with the unstabilized D-loop conformation that we were unable to resolve by classification (Supplementary Fig. 7c, d). There is also a shift in residues 51–53 outside of the D-loop, which is not seen in other structures. Here, the volume suggests that substitution Cys53, not found in skeletal actin or *P. falciparum* Act1, may have become oxidized, possibly leading to the change (Supplementary Fig. 7f).

The C-terminus of the protein is unresolved past residue His372 in the jasplakinolide-bound structure (Supplementary Fig. 7g). More subtle shifts occur among the subdomains within one protomer. The overall effect is for residues 139–144 and 169–173 in subdomain 2 and residues 352–370 in subdomain 1 to pinch the D-loop in its binding pocket, likely stabilizing its position (Fig. 5e), which suggests that the *Tg*Act1 D-loop can adopt at least two conformations within the filament.

The *Tg*Act1 protomers from jasplakinolide-bound filaments show high structural similarity to protomers from jasplakinolide-bound, *P. falciparum* Act1 filaments, with a Cα RMSD value of 0.6 (residues 6–372, PDB ID 6tu4)[52,53]. The helical symmetries of filaments from *Tg*Act1 and *P. falciparum* Act1 are very similar (*P. falciparum* Act1: 28.4 Å rise and −167.6° twist)[53]. The D-loop of the jasplakinolide-bound filament adopts a conformation nearly identical to that of jasplakinolide-bound *P. falciparum* Act1 and skeletal actin loops (Fig. 5d, f). Cα RMSD values comparing the D-loops demonstrate the overlap: 0.8 and 1.2 comparing jasplakinolide-bound *Tg*Act1 to jasplakinolide-bound *P. falciparum* Act1 and native (i.e., unstabilized) chicken skeletal actin, respectively.

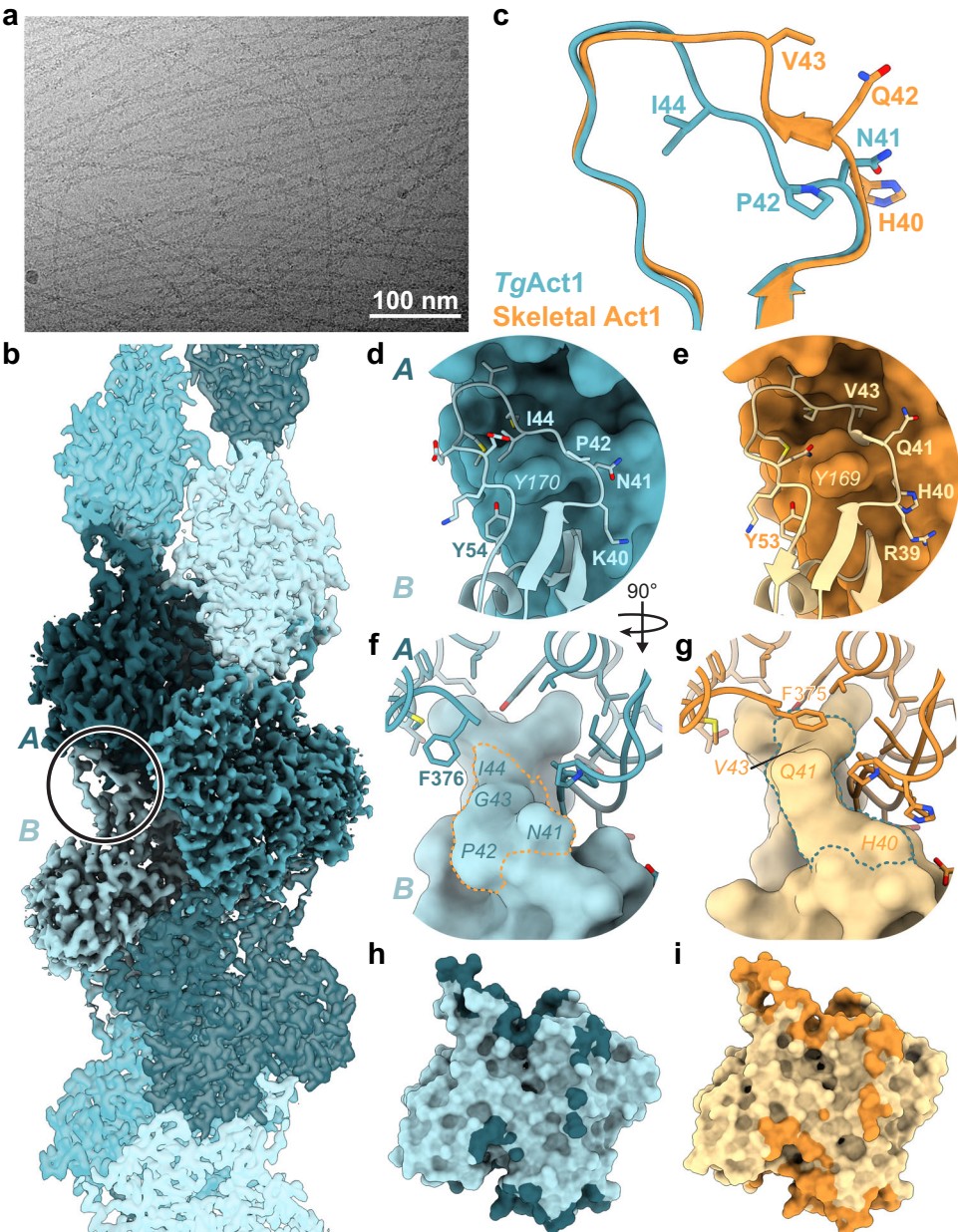

**Fig. 4 | Comparison of unstabilized *Tg*Act1 filament to skeletal actin filaments.**
**a** Micrograph of *Tg*Act1 filaments in the presence of 1 mM MgATP; see Table 2 for dataset statistics. **b** Reconstruction of *Tg*Act1 filaments in the presence of 1 mM MgATP; protomers in shades of blue. The circle indicates the location of the ᴅ-loop in one protomer. **c** Overlay of ᴅ-loops from unstabilized *Tg*Act1 filament model (*Toxo* Act1, blue) and the skeletal actin filament 8d13 (orange). **d, e** View of the ᴅ-loop (ribbon and sticks) and binding pocket (surface) from *Tg*Act1 (**d**, blue) and skeletal actin (**e**, orange). ᴅ-loop residues within 5 Å of the pocket are shown as sticks. **f, g** View of the ᴅ-loop (surface) and binding pocket (ribbon and sticks) from *Tg*Act1 (**f**, blue) and skeletal actin (**g**, orange). Residues within 5 Å of the ᴅ-loop are shown as sticks, except for *Tg*Act1 F376, which is shown for illustrative purposes only. Dotted lines indicate the locations of ᴅ-loop residues 41–44 (*Tg*Act1) and 40–43 (skeletal actin). **h, i** Surface representation of a protomer of *Tg*Act1 (**h**, blue) and skeletal actin (**i**, orange), with the shaded regions showing the buried surface area for each protomer.

A comparison of jasplakinolide-bound *Tg*Act1 and *P. falciparum* Act1 reveals a few differences. Alignment of the protomers shows small changes in the positioning of the C-terminal backbone when comparing *Tg*Act1 and *P. falciparum* Act1 (Fig. 5g). The unresolved C-terminus and the shift in the backbone position change the hydrogen bond network that C-terminal residues Ser369 and His372 make with residues Lys114, Glu117, and Arg118. As mentioned above, the ᴅ-loops contain the same residues; however, there are three amino acid substitutions immediately outside of the ᴅ-loop. In addition to the Cys53 substitution mentioned above, *Tg*Act1 contains Lys38, and Tyr54, while *P. falciparum* Act1 contains Arg38 and Phe54 (Ext Data Fig.

8). Overall, these are conservative substitutions; Lys38 and Tyr54 adopt similar positions to Arg38 and Phe54.

Neither actin isoform from *P. falciparum* shows large differences in the positioning of the ᴅ-loop when compared to skeletal actin filaments (Fig. 5d, f)[25,52,54]. For *P. falciparum* Act2, the substitution of asparagine for the glycine at position 42 might facilitate a more stable filament. In contrast, biochemical work on *P. falciparum* Act1 suggests that the ᴅ-loop contributes to the instability of the filaments; substitution of the ᴅ-loop of *P. falciparum* Act1 with the canonical human actin ᴅ-loop decreased the critical concentration for assembly of *P. falciparum* Act1[22,25]. Because of the high identity of *P. falciparum* Act1

**Table 2 | Cryo-EM data collection, refinement, and validation statistics**

| | TgAct1 | | TgAct1 + Jasplakinolide | |
|---|---|---|---|---|
| | **Filament** | **3x Protomers** | **Filament** | **3x Protomers** |
| Sample composition | | | | |
| Protein | TgAct1 | | TgAct1 | |
| Prot. Conc | 30 µM | | 14 µM, 60 µM | |
| Buffer | 5 mM Tris, pH 8 | | 5 mM Tris, pH 8 | |
| | 0.2 mM CaCl2 | | 0.2 mM CaCl2 | |
| | 200 µM DTT | | 200 µM DTT | |
| | 200 µM NaATP | | 200 µM NaATP | |
| | 50 mM KCl | | 50 mM KCl | |
| | 2 mM MgCl2 | | 2 mM MgCl2 | |
| Ligand added | 1 mM MgATP | | 0.1 mM MgATP 33 µM Jasplakinolide | |
| Data Collection | | | | |
| Microscope | Krios | | Glacios | |
| Camera | K3 | | K3 | |
| Number of Micro-graphs (total) | 3244 | | 5334 | |
| Nominal magnification | 105,000 X | | 45,000 X | |
| Voltage | 300 kV | | 200 kV | |
| Electron Fluence | 60 e⁻/Å² | | 50 e⁻/Å² | |
| Pixel Size | 0.842 Å | | 0.89 Å | |
| Defocus range | −0.70 to −1.85 µm | | −0.75 to −2.5 µm | |
| Map | | | | |
| Centering | Filament | 3x Protomers | Filament | 3x Protomers |
| Number of parti-cles (final) | 2,248,567 | 2,246,540 | 2,682,453 | 1,351,186 |
| Symmetry Imposed | Helical | None | Helical | None |
| Rise (Å) | 28.1 | | 28.1 | |
| Twist (°) | −167 | | −167 | |
| Map resolution (Masked) $FSC_{0.143}$ | 2.7 | 2.6 | 3.3 | 3.0 |
| B-factor | 131.4 | 105.4 | 193.9 | 142.7 |
| Density modified resolution (masked) $Ref_{0.5}$ | 2.5 | 2.5 | 3.2 | 3.0 |
| Model | | | | |
| Starting model | 6TU4 | | 8TRM | |
| Composition | | | | |
| Protein residues | 1113 | | 1101 | |
| Ligands | 3 ADP | | 3 ADP | |
| | 3 $Mg^{2+}$ | | 3 $Mg^{2+}$ | |
| | | | 3 Jasplakinolide | |
| Waters | 30 | | | |
| R.M.S. deviations | | | | |
| Bond lengths (Å) | 0.004 | | 0.005 | |
| Bond angles (°) | 1.022 | | 1.048 | |
| Validation | | | | |
| Molprobity score | 1.24 | | 1.27 | |
| Clash score | 2.81 | | 4.46 | |
| Rotamer outliers (%) | 1.68 | | 0.64 | |
| B-factors (min/max/mean) | | | | |
| Protein | 11/85/37 | | 19/111/50 | |
| Ligand | 11/39/28 | | 29/125/55 | |
| Water | 17/38/29 | | | |

**Table 2 (continued) | Cryo-EM data collection, refinement, and validation statistics**

| | TgAct1 | | TgAct1 + Jasplakinolide | |
|---|---|---|---|---|
| | **Filament** | **3x Protomers** | **Filament** | **3x Protomers** |
| Ramachandran plot | | | | |
| Favored (%) | 98.64 | | 97.81 | |
| Allowed (%) | 1.36 | | 2.19 | |
| Outliers (%) | | | | |
| EMDB ID | EMD-41583 | | EMD-41584 | |
| PDB ID | 8TRM | | 8TRN | |

to TgAct1, it appears possible that in unstabilized filaments the *P. falciparum* Act1 ᴅ-loop could adopt the same position as the unstabilized TgAct1 ᴅ-loop.

## Discussion

In this study, we show that TgAct1 is capable of polymerizing into long filaments in the absence of assembly factors, stabilizers, or the actin chromobody, which is currently the only actin probe capable of detecting actin filaments in Apicomplexan parasites. Unlike skeletal actin, TgAct1 filaments rapidly treadmill at concentrations of actin above 12 µM (Fig. 6). This unusual feature has also been observed for *P. falciparum* Act1[22] which shares 93% sequence identity to TgAct1 and thus, appears to be a conserved feature of Apicomplexican actin.

The structural basis for rapid disassembly is likely explained by the conformation and dynamics of the ᴅ-loop, although both the less ordered water network and the nucleotide sensing residues may indirectly influence the ᴅ-loop positioning and disassembly. Our structures reveal at least two ᴅ-loop conformations for TgAct1; while we have modeled one conformation for each structure, the volumes suggest some mixed occupancy. In the jasplakinolide-stabilized structure, the ᴅ-loop is inserted into the binding pocket, whereas in the absence of stabilizing agents, the loop is partially out of the binding pocket. While speculative, one possibility is that the ᴅ-loop could progress through different conformations as the filament ages, with the loop making the full set of contacts during filament assembly and then losing contacts during the aging process, decreasing the number of longitudinal interactions and speeding disassembly. Given the role of the ᴅ-loop in regulating F-actin stability through longitudinal contacts, we interpret the reduction of longitudinal interactions in the unstabilized structure to account for the reduced filament stability and high dissociation constants at the pointed-end that drive treadmilling (Fig. 6, inset). Further support for this idea comes from data showing that replacing the *P. falciparum* Act1 D-loop with the mammalian sequence has a stabilizing effect on the filament[25] accompanied by a ~5-fold reduction in the critical concentration[22]. However, despite the known importance of the ᴅ-loop in regulating *P. falciparum* Act1 filament stability, previous structures of *P. falciparum* Act1 stabilized with jasplakinolide observed minimal differences in the position of the D-loop of *P. falciparum* Act1 relative to skeletal muscle actin[52].

To support rapid treadmilling, TgAct1 monomers must quickly release ADP and recharge with ATP in the absence of nucleotide exchange factors. Exploration of the hinge regions involved in the F to G transition did not reveal any substantive differences between TgAct1 and skeletal actin. There is only one substitution within the nucleotide-binding pocket between skeletal muscle actin and TgAct1: a leucine (Leu16) is substituted with an asparagine (Asn17) in TgAct1 (Supplementary Figs. 8, 9h). *P. falciparum* Act1, like TgAct1, also contains this substitution[25]. Comparison of the gelsolin-bound actin structures of *P. falciparum* Act1 and skeletal actin[55,56] to the stabilized filament structure of *P. falciparum* Act1 and the unstabilized filament structure of

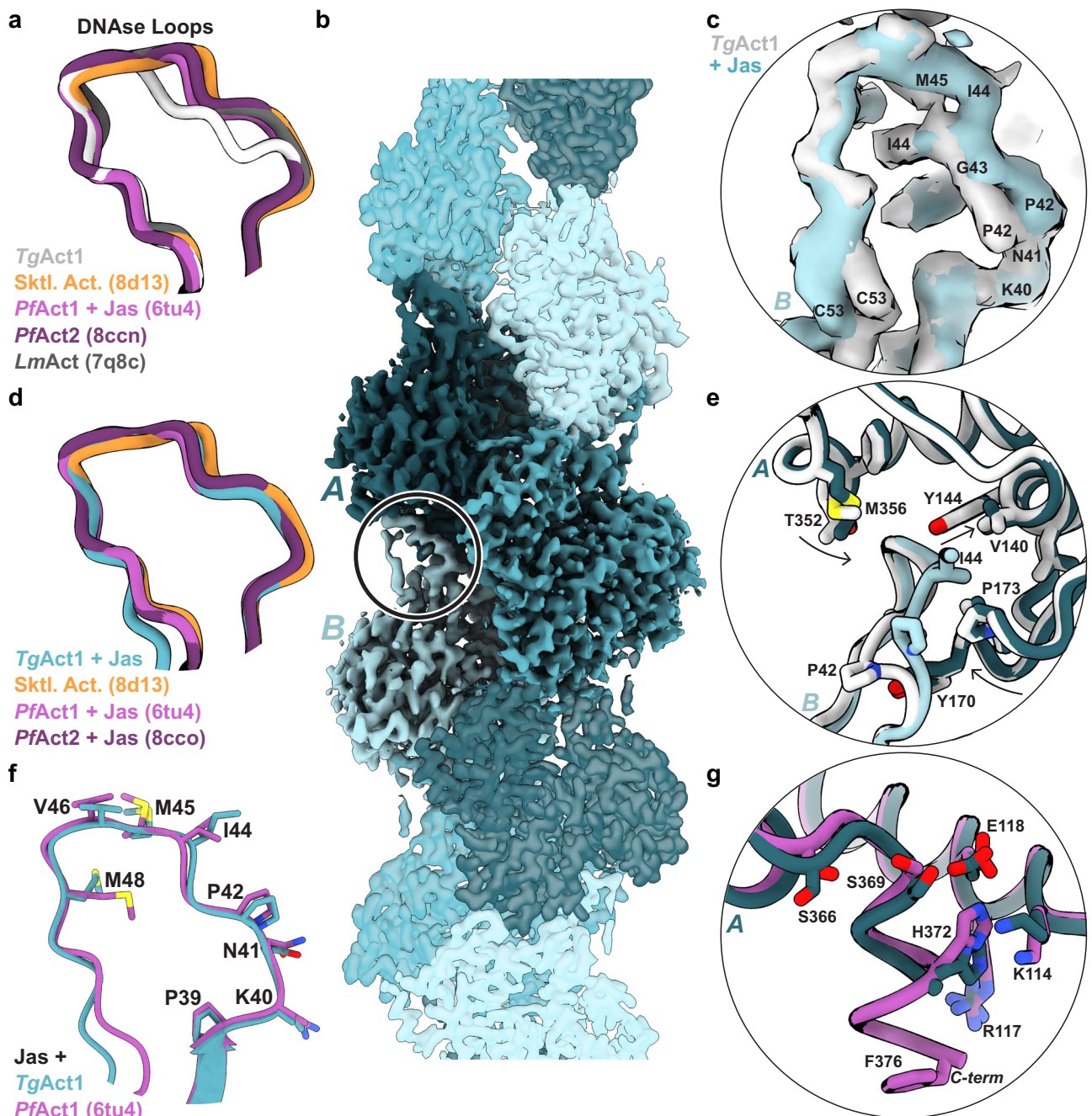

**Fig. 5 | Comparison of stabilized and unstabilized *Tg*Act1 filaments to actin filaments from other species. a** The D-loops from unstabilized *Tg*Act1 (gray), chicken skeletal actin (Sktl. Act., orange, PDB ID 8d13), *P. falciparum* Act1 + jasplakinolide (*Pf*Act1 + Jas, magenta, PDB ID 6tu4), *P. falciparum* Act2 (*Pf*Act2, purple, PDB ID 8ccn), and *L. major* Act (*Lm*Act, dark gray, PDB ID 7q8c). **b** Reconstruction of *Tg*Act1 filaments in the presence of 33 μM jasplakinolide and 0.1 mM MgATP; protomers in shades of blue. The circle indicates the location of the D-loop in panel **c** and the approximate locations of the ribbon diagrams in **e** and **g**. **c** Overlay of volume from the D-loops of *Tg*Act1 + jasplakinolide (blue) and unstabilized *Tg*Act1 (gray). **d** The D-loops from *Tg*Act1 + jasplakinolide (blue), chicken skeletal actin (orange, PDB ID 8d13), *P. falciparum* Act1 + jasplakinolide (magenta, PDB ID 6tu4), and *P. falciparum* Act2 + jasplakinolide (purple, PDB ID 8cco). **e** Overlay of the D-loop and binding pocket from unstabilized *Tg*Act1 filaments (gray) and *Tg*Act1 + jasplakinolide filaments (blue). The backbone is shown as a ribbon with select residues shown as sticks. **f** Overlay of ribbon diagrams showing D-loop residues 39–48 from *Tg*Act1 + jasplakinolide and *P. falciparum* Act1 + jasplakinolide (magenta, PDB ID 6tu4). **g** Overlay of ribbon diagram comparing the C-termini from *Tg*Act1 + jasplakinolide (dark teal) and *P. falciparum* Act1 + jasplakinolide (magenta, PDB ID 6tu4).

*Tg*Act1 suggests how the residue might affect the nucleotide exchange rate. The G-actin structure of *P. falciparum* Act1 shows Asn17 hydrogen bonding with the backbone carbonyl of Phe34 in the adjacent beta-strand. For this change to occur, the hydrogen bond with the alpha phosphate in the filament protomer must be broken and the gamma carbon of Asn17 must move 1.6 Å toward the backbone carbonyl of Phe34. In contrast, the gamma carbon of hydrophobic Leu16 only moves half that distance (Supplementary Fig. 9i). The substitution of the asparagine for the leucine and the associated shifts in G-actin unbury part of the nucleotide, and decrease the hydrophobic character while widening a portion of the binding pocket, potentially facilitating nucleotide exchange.

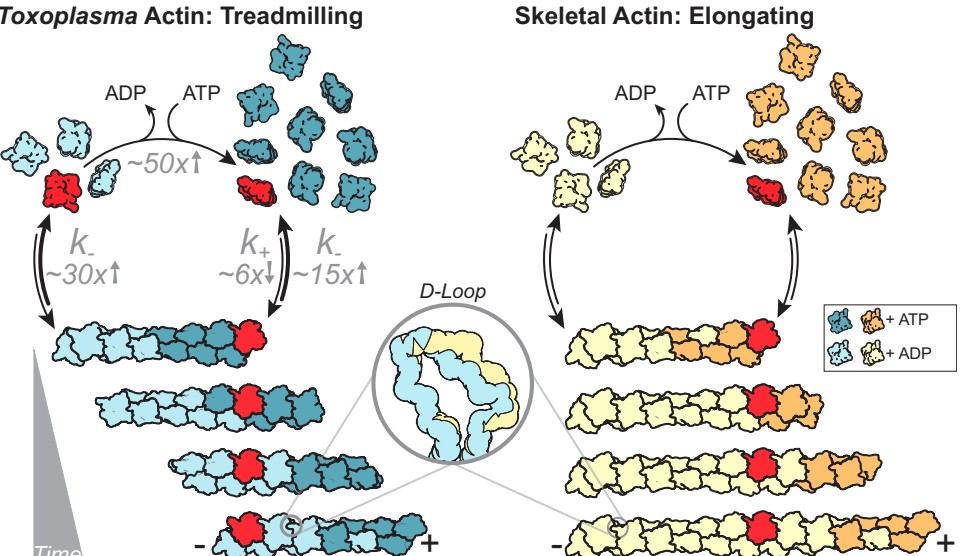

**Fig. 6 | The filament properties of *Toxoplasma* actin versus skeletal actin.** A low barbed-end assembly rate and a high pointed-end disassembly rate leads to a high critical concentration for *Tg*Act1 relative to skeletal actin—an effect mediated in part by changes within the D-loop of *Tg*Act1. Paired with the ability to rapidly exchange nucleotide, actin filaments in *T. gondii* rapidly treadmill at concentrations where skeletal actin filaments elongate. Change in rates of *Tg*Act1 assembly, disassembly, and nucleotide exchange compared to skeletal muscle actin are indicated in gray text. k., disassembly: subunits·sec$^{-1}$; k$_+$, assembly: subunits·μM$^{-1}$·s$^{-1}$.

This work challenges the idea that *Tg*Act1 forms only short unstable filaments. We consistently observe filaments longer than 20 μm using fluorescent microscopy, and filaments longer than 1 μm that span the micrographs in negative stain electron microscopy. In addition, bulk ATPase assays indicate that 24 μM *Tg*Act1 monomers treadmill into and out of the filament three times per hour. Given the dissociation constant from the pointed-end (11 sec$^{-1}$), the average filament length under these conditions is estimated to be ~36 μm. Previous methods for measuring *Tg*Act1 filament length were either indirect, using centrifugation, or by negative stain electron microscopy of filaments near their critical concentration, where low availability of monomers limits their length[49]. It is unclear whether the N-terminal HIS tag used in previous studies influences *Tg*Act1 or *P. falciparum* Act1 filament stability and length.

While there are differences in the reported critical concentration for Apicomplexan actins, our results are largely consistent with the high critical concentration obtained by Lu et al. for *P. falciparum* Act1[22]. Previous experiments that found low critical concentrations for *Tg*Act1[23] or *P. falciparum* Act1[24] were performed over time courses many hours in length. Our ATPase results indicate that unlike skeletal actin, *Tg*Act1 will be completely converted to MgADP (Fig. 2c) and long filaments will be lost over these long time periods because ADP-actin has a tenfold increased critical concentration compared to ATP-actin[22,31]. In addition, monitoring the polymerization of apicomplexican actin overnight is also problematic because small oligomers form when not stabilized by the addition of ammonium acetate[24], which can complicate the interpretation of light scattering or pyrene fluorescence.

In vivo visualization of actin in *T. gondii* using the actin chromobody has revealed a vast network of highly dynamic actin that is well supported by its in vitro properties[12]. Because the concentration of *Tg*Act1 in the cell is near the critical concentration for the pointed-end, its assembly and disassembly may be locally fine-tuned by actin sequestering proteins which could function to drive the dynamics of F-actin in *T. gondii*. Apicomplexan parasites have an unusually limited repertoire of actin-binding proteins[29]. *T. gondii* encodes a single gene of ADF and profilin, and three formins. In mammalian cells, profilin binds to G-actin and plays a dual function in nucleotide exchange and enhancement of formin-mediated actin polymerization[57–59].

Considering our finding that *Tg*Act1 has a fast nucleotide exchange rate, the function of profilin may be restricted to enhancing actin polymerization from one of the three formins in the cell.

The actin-depolymerizing factor (ADF)/cofilin is a family of proteins that regulate the turnover of actin networks in the cell[60–63]. ADF/cofilin proteins typically bind F-actin and destabilize filament networks by severing actin filaments and accelerating depolymerization from the pointed-end. ADF is essential to the survival of *T. gondii* and its depletion promotes the assembly of F-actin structures in the cell[7,17]. Interestingly, Apicomplexan ADF lacks the residues for binding F-actin and therefore shows very weak severing activities. Instead, it exclusively binds and sequesters G-actin[7,64]. Severing activities was likely lost because of the high disassembly rates of *Tg*Act1, which did not necessitate severing for fast disassembly. Thus, ADF functions primarily in *T. gondii* to regulate the pool of available actin monomers.

A key property of *Tg*Act1 that makes it primed for rapid disassembly is the ~30-fold higher disassembly rate for the pointed-end compared to skeletal actin. Thus, below the critical concentration, *Tg*Act1 filaments will depolymerize ~30 times faster than conventional actin. Considering our findings that the cellular concentration of *Tg*Act1 is less than twofold from the critical concentration of pointed-end, small changes in the expression of ADF, or its activation through post-translational modification may allow for rapid transitioning from filament to monomer without the need for actin severing proteins or disassembly factors.

Why does Apicomplexican actin have an unusually high critical concentration? This property would seemingly require the expression of roughly ten times more actin and regulatory proteins to generate filaments which is energetically wasteful. We propose that the high critical concentration is a functional consequence of the high monomer disassembly rates. While more actin expression is required for filament growth, the fast disassembly rates allow the parasite to quickly disassemble the actin network with a minimal set of actin regulatory factors. Such a mechanism could have evolved due to constraints of the lytic cycle on parasite survival. Intracellular parasites contain an extensive and dynamic actin network. When calcium ionophore is used to trigger parasite egress the network rapidly disassembles in less than 60 s, prior to the onset of parasite motility[17]. During motility and egress, actin is enriched at the parasites basal

end[19]. Thus, parasite motility and the transition from the intracellular to extracellular states (and vice versa), necessitate fast reorganization of the actin network and transitions between F and G-actin.

Due to its ubiquity across the domains of life, actin serves as a model for the evolutionary tuning of the dynamic properties of a conserved polymer. In general, eukaryotic actin is highly conserved at the sequence level and in the overall structure of the filament, and achieves functional diversity in part through interactions with a host of regulatory proteins that tune its dynamic properties. Bacterial actins, on the other hand, typically have many fewer interacting partners, which has allowed evolutionary diversification to yield different dynamic properties through sequence changes that introduce significant changes to the filament architecture[65]. *Tg*Act1 sits between these two extremes, demonstrating that smaller evolutionary changes can give rise to new dynamic properties without large changes to the overall filament structure.

## Methods

### DNA expression constructs

*Tg*Act1 (ToxoDB: TgME49_209030) was amplified from *T. gondii* cDNA with the primers 5′-acaatcacgcggccgccaccatggcggatgaagaagtgc-3′ and 5′-gcccgagcctcccgagctagcgaagcacttgcggtggacg-3′ and fused as its C-terminus to a 14 amino acid GS linker sequence followed by a β-thymosin–6xHIS tag and inserted into pFastBac (Thermo Fisher Scientific) behind the p10 promoter for expression in *Sf*9 cells. The β-thymosin fusion strategy was previously used for the expression of *Plasmodium falciparum* and *Dictyostelium* actin, which shares 95 and 85% identity to *Tg*Act1 respectively[22,26,27]. The Actin-chromobody (ChromoTek Inc., Hauppauge, NY) containing a C-terminal EmeraldFP–6xHIS tag was cloned into pET22b (Novagen) for expression in *E. coli* BL21(DE3)[17].

### Protein expression and purification

Expression and purification of untagged *Tg*Act1 was adapted from previous methods[22,26]. Two billion *Sf*9 cells were infected with a virus and grown for 3 days at 27 °C. Cells were harvested by centrifugation at 2000 RPM and resuspended in 50 mL lysis buffer (10 mM HEPES, pH 8.0, 0.25 mM CaCl₂, 0.3 M NaCl, 0.5 mM DTT, 0.5 mM Na₂ATP, 5 μg/mL leupeptin, 1x cOmplete ULTRA Tablets (Roche, 05892791001). Cells were lysed by sonication in an ice water bath and then clarified at 250,000×*g* for 35 min. The supernatant was then added to 3 mL of HIS Select Nickel affinity gel (Sigma, P6611) equilibrated in wash buffer (10 mM HEPES, pH 8.0, 0.25 mM CaCl₂, 0.3 M NaCl, 0.5 mM DTT, 0.25 mM Na₂ATP) for 45 min at 4 °C. The affinity gel was then lightly sedimented at 4 °C, 600 RPM and added to a column and washed with 15-bed volumes of wash buffer followed by six-bed volumes of wash buffer containing 10 mM Imidazole (pH 8). The protein was eluted with a wash buffer containing 200 mM Imidazole, pH 8. Peak fractions were concentrated to 2 mL Amicon Ultra-15 (Millipore, 2022-08-01) and dialyzed against 1 L G-buffer overnight (5 mM Tris, pH 8.2, 0.25 mM CaCl₂, 0.5 mM DTT, 0.25 mM Na₂ATP, 1 μg/mL leupeptin, 4 °C). Protein was divided into 150 μL aliquots, snap-frozen in liquid nitrogen and stored at −80 °C. Small-scale chymotrypsin digests were performed to determine the optimal molar ratios of chymotrypsin:*Tg*Act1–β-thymosin–HIS for removal of the β-thymosin–HIS tag. Volumes of 20 μL of actin were incubated with chymotrypsin (Sigma, c3142) at the indicated molar ratio (Supplementary Fig. 1a) in a 27 °C water bath for 15 min and then quenched with 0.5 mM PMSF (Roche, 10837091001). Samples were analyzed on a Coomassie-stained SDS-PAGE PAGE gel. Once an optimal molar ratio was determined, the remaining prep was digested, quenched with PMSF, clarified 400,000 × *g* for 30 min and applied to a Resource Q (GE Healthcare, 17117701) ion-exchange column equilibrated with G-buffer. *Tg*Act1 was separated from the cleaved tag by applying a 0–0.5 M NaCl salt gradient in G-buffer. Fractions containing purified tagless *Tg*Act1 were concentrated to

0.5 mL and dialyzed overnight against G-buffer containing 0.2 M ammonium acetate, pH 8, 4 °C, which prevents spontaneous actin oligomerization[22,24]. The following day, the sample was clarified at 400,000×*g* for 30 min and applied to a Superdex 200 Increase 10/300 GL (GE Healthcare, 28990944) gel filtration column equilibrated in G-buffer containing 0.2 M ammonium acetate. Fractions corresponding to monomeric actin were pooled, concentrated in an Amicon Ultra − 4 (Millipore, 2021-12-01), snap-frozen in liquid nitrogen and stored at −80 °C.

The actin chromobody was purified from *E. coli*, BL21(DE3) (New England Biolabs C2527). Transfected cells were grown to mid-log (OD = 0.7), induced with 0.4 mM IPTG (Roche, 10724815001), and grown overnight at 16 °C. Cells were harvested by centrifugation at 3500 RPM and resuspended in 40 mL lysis buffer (10 mM NaPO₄, 300 mM NaCl, 0.5% glycerol, 7% sucrose, pH 7.4 at 22 °C, 0.5% NP40, 0.5 mM PMSF, 1x protease inhibitor tablets (Pierce, A32965). Lysozyme (Sigma, L6876) was added to 3 mg/mL and incubated at 4 °C for 30 min. The cell lysate was then sonicated in an ice water bath and then clarified at 250,000×*g* for 35 min. The supernatant was then added to 3 mL of HIS Select Nickel affinity gel (Sigma, P6611) equilibrated in wash buffer (10 mM NaPO₄, 300 mM NaCl, 0.5% glycerol, pH 7.4 at 22 °C, 0.5 mM DTT) for 45 min at 4 °C. The affinity gel was passed over a column and washed with 20 bed volumes of wash buffer followed by 10 bed volumes of wash buffer containing 10 mM Imidazole (pH 7.4). The protein was eluted with a wash buffer containing 200 mM Imidazole, pH 7.4. Peak fractions were concentrated to 3 mL and dialyzed against 0.2 L glycerol storage buffer (25 mM Imidazole, pH 7.4, 300 mM NaCl, 50% glycerol, 1 mM DTT). The final protein concentration was determined using a Bradford reagent (Thermo Scientific, 23238).

### In vitro actin growth assay

*Tg*Act1 was thawed and clarified at 400,000×*g* for 30 min. The storage buffer was exchanged with G-buffer using a Zeba spin-desalting column (Thermo Fisher, 89882) and the actin was stored on ice for up to 4 h. The concentration of the actin was determined by absorbance at 280 nm (1 OD = 38.5 μM). It should be noted that determining actin concentration using Bradford gives a measurement approximately twofold less than that measured by absorbance which could account in part for ~2-fold difference in the calculated $C_c$ between *Plasmodium* actin[22] and *Toxoplasma* actin (this study). The visualization of actin growth in vitro was adapted from previous studies on *Plasmodium* actin[22,26]. Flow chambers were bound to minimal densities of NEM-treated muscle myosin, which is ATP-insensitive and functions to keep filaments in the imaging plane. The flow cells were then rinsed with ice-cold buffer B (25 mM Imidazole, pH 7.4, 50 mM KCl, 2.5 mM MgCl₂, 1 mM EGTA, 10 mM DTT), and blocked with buffer B containing 5 mg/mL BSA and 1% Pluronic F127 (Invitrogen, P6866). The indicated concentrations of G-actin diluted in G-buffer was mixed 1:1 with an equal volume of 2x polymerization buffer (50 mM Imidazole, pH 7.4, 5% methylcellulose, 100 mM KCl, 5 mM MgCl₂, 2 mM EGTA, 20 mM DTT, 5 mg/mL BSA, 1% Pluronic F127, 5 mM MgATP, 50 nM actin chromobody-EmeraldFP, and an oxygen scavenging system (0.13 mg/mL glucose oxidase, 50 μg/mL catalase, and 3 mg/mL glucose), and passed twice through the flow chamber which was then immediately sealed with nail polish. Actin polymerization which was visualized by low concentrations of actin chromobody was imaged at 37 °C in epifluorescence using a DeltaVision Elite microscope (Cytiva) built on an Olympus base with a 100 × 1.39 NA objective and definite focus system. Images were acquired every 5 s for 5 to 30 min with a scientific CMOS camera and DV Insight solid-state illumination module.

### Epifluorescence microscopy data processing

Movies of actin growth were analyzed using the ImageJ plugin MTrackJ[66]. The rate of shrinkage or growth for each filament was

determined by measuring the change in filament length at each end over the length of the 5-min movie. The length of *Tg*Act1 or skeletal actin was converted to the number of actin subunits in Excel with 2.76 or 2.74 nm length increase per subunit respectively[25]. Pauses in depolymerization at the pointed-end were excluded from rate determinations. Individual measurements were imported into GraphPad Prism (GraphPad Software ver. 10.0.0, Boston, Massachusetts USA) for statistical analysis and graphing. Kymographs of actin dynamics were generated using the ImageJ plugin multiplekymograph. Error measurements associated with critical concentration measurements represent the standard deviation of three independent preparations of *Tg*Act1.

## Measuring the rate constant of ethenoATP dissociation from actin monomers

For the preparation of ethenoATP (εATP)-bound actin, *Tg*Act1 or skeletal muscle actin monomers were clarified at 400,000×*g* for 30 min and exchanged into G-buffer as described above. Actin was diluted to 20 μM in 200 μL and bound calcium was exchanged for magnesium by simultaneously mixing with 0.2 mM EGTA and 80 μM MgCl$_2$ followed by incubating on ice for 5 min. Free ATP was depleted from the solution with 10 μL DOWEX AGX1 resin (BioRad 143-2445) that was washed and equilibrated in 2 mM Tris (pH 8.0). The mixture was continuously inverted for 2 min and centrifuged at 13,800 ×*g* for 1 min at room temperature to remove the resin and bound nucleotide. To the supernatant, 50 μM εATP (final concentration, Jena Bioscience, NU-1103S) was added and the mixture was incubated on ice for 30 min. To measure the nucleotide exchange rate constant, 150 μL εATP-actin was added to a 96-well plate and 50 μL of 2 mM MgATP was added using an auto-injector in a SpectraMax® i3x plate reader. The plate was then simultaneously mixed and measured for fluorescence at 410 nm resulting in a delay time of 0.2 s after injection. Fluorescence readings were taken at 0.2 s intervals for 5 min. Time courses of fluorescence change were fitted to single exponential decay in the form of $FI_{obs}(t) = Amp(1 - e^{kt}) + FI_0(t_0)$, where $FI_{obs}(t)$ is the observed fluorescence intensity signal at time $t$, $Amp$ is the amplitude of the fluorescence signal change (negative value for an exponential decay), $FI_0(t_0)$ is the initial fluorescence intensity signal at time $t = 0$, and $k$ is the observed εATP dissociation rate constant[67], using GraphPad Prism (GraphPad Software ver. 10.0.0 (Boston, Massachusetts USA). Because nucleotide dissociation is rate-limiting and nucleotide-binding is far more rapid, the observed rate constant of the best-fit yields the nucleotide dissociation rate constant. Uncertainties associated with the average nucleotide dissociation rate constants are represented as the standard deviation of the mean for three independent experiments.

## Actin Pi release assay

*Tg*Act1 or skeletal actin was clarified at 400,000 ×*g* for 30 min and exchanged into fresh G-buffer as described above. The actin solution was warmed for 30 s in a 37 °C water bath and induced for polymerization by adding 10x polymerization buffer (250 mM Imidazole, pH 7.4, 500 mM KCl, 20 mM MgCl$_2$, 10 mM EGTA, 2 mM MgATP, 10 mM DTT) to a 1x final concentration at 37 °C. The amount of Pi released from actin was measured by absorbance at 360 nm every 20 s for 30 min to 1 h using an EnzChek phosphate assay kit (Thermo Fisher, E12020)[68,69] in a GENESYS™ 10 UV-Vis Spectrophotometer (Thermo Scientific) maintained at 37 °C.

## Western blotting to determine the cellular actin concentration

Cells were syringe released, counted, and harvested by centrifugation at 3500 r.p.m. for 4 min in a benchtop centrifuge. Cells were washed in PBS, and boiled for 7 min in SDS-PAGE loading buffer. Cell lysates equivalent to $2.5 \times 10^6$ and $5 \times 10^6$ cells/lane were run on SDS-PAGE next to known amounts of purified *Tg*Act1 ranging from 50 to 300 ng as

determined by OD280. Protein was then transferred to a nitrocellulose membrane, blocked with blocking buffer (2.5% non-fat milk in PBS), and incubated with a 1/1000 dilution in blocking buffer of rabbit anti-actin-antibody (Gift from Marcus Meissner) which binds *Tg*Act1[8]. The blot was then washed, incubated with a 1/3000 dilution of goat anti-rabbit secondary antibody in blocking buffer conjugated to IRDye® 680RD (LI-COR), washed, and imaged on a Typhoon imager (Cytiva). Densitometry analysis was performed using ImageJ to generate a standard curve of intensities from known actin concentrations and the ng amount of actin per lane was determined by a standard fit to the curve. The number of parasites per lane and cell volume determined previously[70] was used to calculate the cellular actin concentration.

## Preparation of rhodamine-labeled TgAct1 filaments

*Tg*Act1 was thawed and desalted twice into a storage buffer (5 mM Tris, pH 8.2, 0.2 M ammonium acetate, 0.25 mM CaCl$_2$, 0.5 mM DTT, 0.25 mM Na$_2$ATP, 4 °C) lacking DTT. The concentration of actin was determined by OD280 and Tetramethylrhodamine-5-Iodoacetamide Dihydroiodide (5-TMRIA) (Fisher Scientific, T6006) was added at a 1:1 molar ratio and incubated at 4 °C overnight. The following day, 1 mM DTT was added to the mixture, clarified 400,000 ×*g* for 30 min, and applied to a Superdex 200 Increase 10/300 GL (GE Healthcare) gel filtration column equilibrated in a G-buffer containing 0.2 M ammonium acetate. Fractions corresponding to monomeric actin were pooled, concentrated in an Amicon Ultra-15 (Millipore, 2022-08-01), snap-frozen in liquid nitrogen and stored at −80 °C. For generating rhodamine-labeled actin filaments, labeled and unlabeled *Tg*Act1 was clarified at 400,000 ×*g* for 30 min and the concentration was determined by OD280. About 1 μM labeled actin was mixed with 20 μM unlabeled actin (5% labeled) and induced for polymerization by adding 2x polymerization buffer without actin chromobody. The mixture was added to a NEM-myosin-bound and blocked flow chamber and allowed to polymerize for 10 min at 37 °C. The flow chamber was then flushed with polymerization buffer containing 50 nM actin chromobody and imaged using epifluorescence microscopy. The resulting actin filaments which were observed using the actin chromobody, produced sparse punctate rhodamine labeling indicating poor rhodamine-*Tg*Act1 incorporation. The average rhodamine-*Tg*Act1 incorporation was determined by counting rhodamine-labeled monomers and dividing by the number of monomers in the filament which was determined using a 2.74 nm length increase per subunit[25] (Supplementary Fig. 1).

## Negative stain electron microscopy

Frozen *Tg*Act1 aliquots were thawed at room temperature and then placed on ice. Samples were spun at 400,000 × *g* for 30 min at 4 °C and the supernatant was desalted using Zeba spin-desalting column (7 K MWCO, Thermo Fisher, 89882) equilibrated with G-buffer supplemented with 0.2 mM DTT and 0.2 mM NaATP. Polymerization was induced by diluting 10x polymerization buffer to 1x in 60 μM (±10) *Tg*Act1 with additives depending on the condition. For *Tg*Act1 + jasplakinolide: jasplakinolide (Millipore, 420127) was added to a final concentration of 33 μM; for *Tg*Act1 + ATP: 10 mM MgATP was used in the 10x polymerization buffer for a final concentration of 1 mM with *Tg*Act1; for *Tg*Act1 + AMPPNP, Li/Na-AMPPNP was substituted for MgATP in the 10x polymerization buffer for a final concentration of 0.1 mM with *Tg*Act1. Immediately after diluting the polymerization buffer into *Tg*Act1, the samples were placed at 37 °C and incubated for 1 h (*Tg*Act1 + Jas), 45 min (*Tg*Act1 with AMPPNP), or samples were taken at time points (*Tg*Act1 with 1 mM ATP). Samples were diluted tenfold in 1x polymerization buffer or directly applied to glow-discharged continuous carbon film grids. Grids were then washed in ddH$_2$O three times unless specified otherwise and negatively stained using 2% uranyl formate. Samples were imaged using an FEI Morgagni operating at 100 kV and a Gatan Orius CCD camera with the software package

Digital Micrograph (v2.10.1282.0) or using an FEI Tecnai G2 Spirit operating at 120 kV and a Gatan Ultrascan 4000 CCD camera with the software package Leginon (v3.4).

## Cryo-electron microscopy and data processing

*Tg*Act1 filaments were assembled as described above. *Tg*Act1 + jasplakinolide was assembled and stored at 4 °C overnight before grid application. *Tg*Act1 was used immediately after a 20 min incubation at 37 °C. Protein solutions were diluted to the concentrations listed in Data Table 1 using 1x polymerization buffer and immediately applied to glow-discharged, C-flat 2/2 holey-carbon EM grids (Protochips Inc.), blotted from the front (*Tg*Act1 + jasplakinolide) or back (*Tg*Act1) of the grid, and plunge-frozen in liquid ethane using a manual plunging apparatus at room temperature in a dehumidified room. Data collection was performed using an FEI Glacios (equipped with a Gatan K3 Summit direct electron detector operating in CDS mode) and an FEI Titan Krios transmission electron microscope operating at 300 kV (equipped with a Gatan image filter (GIF) and post-GIF Gatan K3 Summit direct electron detector operating in CDS mode) both using the software package Leginon[71] (v3.5).

Movies were aligned, corrected for beam-induced motion, and dose-weighted using the Relion[72] 4.0 implementation of MotionCor2[73] (v1.3.1); CTF parameters were initially estimated using CTFFind4 (v4.1.10)[74]. Motion-corrected micrographs were imported into cryoSPARC (v4) CTF parameters were re-estimated using Patch CTF Estimation. Particles were picked using Filament Tracer[75]. Two rounds of 2D classification were used to remove noise, carbon edges, and poorly aligning particles, and the remaining particles were used in ab initio reconstruction and Helix Refinement. Initial helical parameters of −167° twist and 27 Å rise were used[76] within a search space of 164–170° and 24–30 Å; helical parameters were refined in all Helix Refinements. Additional rounds of Helix Refinement with dynamic masks and Local Refinement with a static mask surrounding three protomers of the filament were completed, interspersed with Global and Local CTF Refinement and map sharpening with resolution estimation (FSC cut-off of 0.143). For the *Tg*Act1 dataset, the particles were signal subtracted to remove filament signal outside of the three central protomers prior to the final Local Refinement. The final refined, unsharpened maps were exported to Phenix (v1.20) where density modification and additional resolution estimation was performed (FSC cutoff of 0.5)[77]. Local resolution estimation was performed using cryoSPARC's Local Resolution Estimation and Local Filtering.

## Model building

PDB ID 6TU4 was used as an initial model for *Tg*Act1. The model was fitted with the correct residues and iteratively refined against both the filament and 3x protomer maps using a combination of ISOLDE[78] (v1.5) in ChimeraX[79] (v1.5), Coot[80] (v0.9.8.1), and phenix.real_space_refine and phenix.validation_cryoem in Phenix (v1.20)[81,82]. Nucleotide-binding site waters were placed by hand, and the water network in the filament was explored using phenix.douse (v1.20)[83]. The *Tg*Act1 model was used as the initial model for the TgAct1 + jasplakinolide dataset.

## Image generation and calculations for micrographs and protein models

Negative stained micrographs were contrast adjusted, rotated, and cropped in Fiji (v2.1.0/1.53c). Cryo-electron micrographs were motion-corrected in Relion's implementation of MotionCorr2 and Gaussian filtered and contrast adjusted in Fiji. We used ChimeraX for protein volume and model visualization, volume alignment (Fit to Model tool or the "fit" command), model alignment (Matchmaker tool or the "align" command), buried surface area calculations ("measure buriedarea" command), hydrophobicity calculations ("hydrophobic coloring" tool), and hydrogen bonds identification ("hbond" tool). All

electron micrograph image, volume, and protein model figures were assembled in Adobe Illustrator CC6 (v26.0.1).

## Reporting summary

Further information on research design is available in the Nature Portfolio Reporting Summary linked to this article.

## Data availability

The cryo-EM maps generated for this manuscript are available from the EMDB (https://www.ebi.ac.uk/emdb/) at the accession codes listed in Table 1 of the manuscript (EMDB IDs: EMDB-41583 (unstabilized), EMDB-41584 (Jas-bound)). The protein models generated for this manuscript are available from the RCSB PDB (https://www.rcsb.org/) at the accession codes listed in Table 1 of the manuscript (PDB IDs: 8TRM (unstabilized), 8TRN (Jas-bound)). Previously deposited structures referenced in this manuscript include: 7bt7, 7r8v, 8a2r, 8a2t, 8d13, 8dmx, 8dmy, 8dnf, 8dnh, 7q8c, 8ccn, 5ooc, 5ood, 6t24, 7pm3, 8cco, 1eqy, 6i4e. Protein sequences were retrieved from Uniprot (https://www.uniprot.org/) at the codes: P53476, Q8I4X0, P68139, and Q8ILW9. Source data are provided with this paper.

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

## Acknowledgements

We thank Kathleen Trybus and Patricia Fagnant (University of Vermont) for providing DNA subcloning constructs and helpful guidance. We thank our colleagues at the University of Connecticut for sharing equipment; Kenneth Campellone and Jim Cole for the use of their AKTA, and Victoria Robinson and Giancarlo Montovano for their assistance with ion-exchange chromatography. We thank the members of the De La Cruz lab (Yale University), Heaslip lab (University of Connecticut) and John Murray (Arizona State University) for thoughtful discussions. We thank Joel Quispe, Sasha Dickinson, and the Arnold and Mabel Beckman Cryo-EM Center at the University of Washington for electron microscope guidance and use. We also thank members of the Kollman group for the feedback provided during cryo-EM data collection and processing. Thanks to Alexander Paredez for initiating this fruitful collaboration. This work was supported by the National Institutes of General Medical Science and Allergy and Infectious Disease of the US National Institutes of Health (grant nos. R35GM136656 to E.M.D.L.C., R35GM149542 and S10OD023476 to J.M.K., 1F32AI145111 to K.L.H., and R35GM138316 to A.T.H.).

## Author contributions

T.E.S. purified TgAct1 and actin chromobody, designed and performed actin growth and biochemical assays. Kinetic assays were designed by E.M.D.L.C. and performed by T.E.S. K.L.H. designed and performed negative stain and cryo-electron microscopy experiments. T.E.S., K.L.H., J.M.K. and A.T.H. performed data analysis and interpretation and wrote the manuscript. All authors edited and revised the manuscript.

## Competing interests

The authors declare no competing interests.
