## [Peer Review File · Nature Communications]

REVIEWER COMMENTS

Reviewer #1 (Remarks to the Author):

Toxoplasma gondii possesses a divergent actin (TgAct1) that is essential for its life cycle and has different polymerisation kinetics to other well-studied mammalian actin isoforms. However, the biochemical and structural basis for this is unknown. Hvorecny and Sladewski et al have presented a robust study combining biochemistry and structural biology to better understand dynamics and assembly of F-actin from *T. gondii*. The authors recombinantly expressed and purified TgAct1 from insect cells and characterized the polymerisation kinetics using labeled actin chromodies, phosphate release and nucleotide exchange assays. This confirmed that TgAct1 polymerises more slowly and has a higher critical concentration for polymerisation than other well-studied actin isoforms. The authors determined single-particle cryo-EM structures of F-TgAct1 stabilised by Jasplakinolide, revealing a D-loop conformation akin to F-actin structures from other organisms. However, unstabilised F-TgAct1 has a more unique D-loop conformation that reduces the buried surface area between subunits relative to Jasp-stabilised F-TgAct1 and skeletal actin from mammals, thereby rationalizing the kinetics described above.

This manuscript is an important advance for understanding both the *T. gondii* lifecycle as well as more general mechanisms for actin polymerisation. It was enjoyable to read and is suitable for publication if the suggestions below can be commented on.

* Recent high-resolution structures of F-actin have shown that subtle changes in conformation are important for filament stability (e.g. Reynolds et al 2022 and Oosterhaart et al 2022). A concern could be that the authors are overlooking these subtle changes and overplaying the importance of the D-loop. Although the resolution here seems slightly lower compared to these previous studies, can the authors resolve any water molecules in the center of the filament or comment on the structures in other regions that might explain a downstream conformational change in the D-loop? For example, conformations of the D-loop might be indirectly influenced by Q137 (or equivalent) and C-terminal residues in response to the nucleotide state.

* Since interpretation of the map and model in the D-loop is important to the claims of this study, the model validation statistics need scrutiny. It seems there are some residues with poor fit in the vicinity of the D-loop for both structures, but especially the Jasp-bound model. The authors acknowledge this in the Results by stating a potential mixed occupancy for the Jasp-bound D-loop, but may oversell the D-loop change between the Jasp-unbound and Jasp-bound state as binary in the Discussion. For example, "Our structures reveal two distinct D-loop conformations for the stabilized vs. unstabilized structure" is slightly inaccurate, since there would be at least two D-loop conformations based on the data. Could the authors acknowledge this in the Discussion and balance this assertion?

* Furthermore on the Discussion regarding the D-loop: “It suggests that the loop makes the full set of contacts during filament assembly and shifts after assembly, decreasing the number of longitudinal interactions.” The jasp-treated conformation might not necessarily reflect filament assembly in the untreated state. This sentence could be toned down to something more speculative.

* The conclusions would be better supported by mutagenesis of the key residues discussed in the D-loop e.g. P42 and I44. Even more compelling would be a loop replacement with residues from mammalian skeletal actin or PfAct1 or PfAct2. But this is likely beyond the scope of this study.

* Figure 4e would benefit from labels for the surrounding residues referred to in the Results text (e.g. residues 139-144, 169-173 and 352-370). The superimposed densities and models in Figure 4c is also quite unclear, perhaps it could be consolidated with 4e and enlarged. Readers can then better appreciate how this D-loop change is accommodated.

* Ext Data Fig 6e is quite unclear. Could it be enlarged and can H372 be labelled in the unstabilized TgAct1 model?

Reviewer #2 (Remarks to the Author):

Actin is an incredibly well conserved polymer that appears to have originated in bacteria prior to specializing into a core component of the eukaryotic cytoskeleton. Some organisms encode divergent actin proteins with unusual biochemical properties. Apicomplexan parasites are an example of this, as they have a divergent actin that has been argued to form incredibly unstable filaments which cannot grow long. In this study, a collaboration between 3 research groups with combined expertise in actin biochemistry, cell biology, and structure, the authors carefully compare the biochemistry of *Toxoplasma* actin to the well-studied vertebrate skeletal actin. In contrast with previous reports, the authors find that TgACT1 is able to form long filaments, though it has the unusual property of treadmilling. The authors quantify the total actin levels in a *Toxoplasma* cell, which they find to be consistent with the in vitro critical concentration for TgACT1 polymer formation. Finally, they provide two high resolution cryo-EM structures of native, untagged TgACT1 with ATP and with the stabilizing agent jasplakinolide.

The paper is well-written and largely easy to follow. All experiments appear to have been carefully and rigorously carried out. While the paper is relatively short and the concepts discussed reasonably straightforward, I feel overall it is of high quality and adds to our understanding both of how apparent minor changes in actin can alter its biochemical properties and helps reconcile previously conflicting reports on Apicomplexan actin when comparing in vitro versus cellular data.

Overall, my comments are relatively minor.

While cryo-EM structures are distinct from the previously published Plasmodium actin (Pospich 2017) in that the actin used is untagged and one structure has only non-analog ATP without any stabilizing agents, the structures of jas-stabilized could and should be directly compared.

Some Figure panels seem out of order as described in the legend which made it a bit confusing following things (e.g. Figure 1).

While the methods section is overall appropriately detailed, the program used for curve fitting and statistical analysis was a bit vague; presumably Graphpad Prism, though this is not stated for fitting (“e.g. fitted to an exponential [in..]”) nor are any statistics provided, e.g. errors of fit, or statistics validating comparisons of curves or values displayed in bar graphs.

Two issues with “This isoform shares only 83% similarity to skeletal alpha actin and mammalian β and γ isoforms (4)”. 1) “percent identity” is not meaningful without noting the substitution matrix used to compare. Normally, the simplest and most robust comparison is percent identity. 2) is it correct to call TgAct1 an “isoform” if there’s only one actin-encoding gene in that organism and you are comparing to different organisms? Ortholog or simply “protein” seems more appropriate here.

It seems appropriate to include the original reference (Mondragon & Frixione 1996) that implicated actin in conoid movement in addition to the more recent paper referenced.

Figure 1e – I see no blue dashed line in this figure.

Reviewer #3 (Remarks to the Author):

In this work, Hvorecny and Sladewski et al. analyzed the kinetics and structure of Toxoplasma actin, which only shares 83% similarity to the skeletal alpha actin, the canonical model for actin biology. This work revealed the extraordinarily high critical concentration of TgAct1, which can be explained by reduced contact sites with the protein backbone around the D-loop. The nucleotide exchange rate of TgAct1 was found to be ~ 50 fold higher than that of the skeletal actin. This can be explained by a single substitution within the nucleotide-binding pocket between the skeletal muscle actin and TgAct1, from a leucine (Leu16) to an asparagine (Asn17). The manuscript is very well written and the results are clearly

presented. It for the first time provides a coherent view of how the basic biochemical properties of TgAct1 is related to its function in vivo. By carefully delineating the optimization of various experimental conditions, this work also effectively clarifies a number of long-standing confusions in the field regarding the polymerization kinetics of TgAct1. I have only a few comments for the authors to address.

1. The effect of Actin Chromobody (Cb) on actin polymerization

The effect of Actin Chromobody (Cb) on polymerization of TgAct1 is somewhat a debatable point. Here the authors showed that the measurements of Cc, disassembly and assembly rate constants are consistent with previously reported values for the skeletal actin. However the effect of Cb on TgAct1 polymerization remains untested. Since Cb is the most commonly used probe for labeling actin in *Toxoplasma*, it would be tremendously helpful for the field if the authors could directly compare the kinetics of TgAct1 polymerization with or without Cb using either light scattering or cryoEM analysis.

2. As this work revealed, much of the prior confusions in the field arose from suboptimal polymerization conditions. I therefore think that Ext Figure 3 should be included as a figure in the main text, as it shows beautifully how the key parameters that need to be tuned for reliable polymerization of TgAct1 can be determined.

3. The legends for panel f and g in Figure 1 are reversed.

4. Page 7, 4th paragraph: "shift after assembly" should be "shift after disassembly"

5. Page 7, 4th paragraph: "The D-loop differences between stabilized and unstabilized TgAct1 filaments helps to reconcile why previous structures of *P. falciparum* Act1 could not fully explain the importance of the D-loop in filament stability."

I don't follow this argument. It seems to me that the importance of the D-loop in filament stability for *P. falciparum* Act1 is still unexplained.

6. Line numbers should be included for the manuscript text.

Response to Reviewers

We thank the reviewers for their reading of the manuscript and their feedback. Below, we have addressed their comments and suggestions. Edits in the manuscript have been identified by page number and quoted in the text below.

REVIEWER COMMENTS:

Reviewer #1 (Remarks to the Author):

Toxoplasma gondii possesses a divergent actin (TgAct1) that is essential for its life cycle and has different polymerisation kinetics to other well-studied mammalian actin isoforms. However, the biochemical and structural basis for this is unknown. Hvorecny and Sladewski et al have presented a robust study combining biochemistry and structural biology to better understand dynamics and assembly of F-actin from T gondii. The authors recombinantly expressed and purified TgAct1 from insect cells and characterized the polymerisation kinetics using labeled actin chromodies, phosphate release and nucleotide exchange assays. This confirmed that TgAct1 polymerises more slowly and has a higher critical concentration for polymerisation than other well-studied actin isoforms. The authors determined single-particle cryo-EM structures of F-TgAct1 stabilised by Jasplakinolide, revealing a D-loop conformation akin to F-actin structures from other organisms. However, unstabilised F-TgAct1 has a more unique D-loop conformation that reduces the buried surface area between subunits relative to Jasp-stabilised F-TgAct1 and skeletal actin from mammals, thereby rationalizing the kinetics described above.

This manuscript is an important advance for understanding both the T gondii lifecycle as well as more general mechanisms for actin polymerisation. It was enjoyable to read and is suitable for publication if the suggestions below can be commented on.

* Recent high-resolution structures of F-actin have shown that subtle changes in conformation are important for filament stability (e.g. Reynolds et al 2022 and Oosterhaart et al 2022). A concern could be that the authors are overlooking these subtle changes and overplaying the importance of the D-loop. Although the resolution here seems slightly lower compared to these previous studies, can the authors resolve any water molecules in the center of the filament or comment on the structures in other regions that might explain a downstream conformational change in the D-loop? For example, conformations of the D-loop might be indirectly influenced by Q137 (or equivalent) and C-terminal residues in response to the nucleotide state.

At 2.5 Å resolution, our reconstruction of unstabilized TgAct1 is slightly less resolved than both Reynolds et al (2022 Nature, PDB ID 8d13; 2.4 Å resolution) and Oosterheert et al (2022 Nature, PDB ID 8a2t; 2.2 Å resolution). To explore the water network in unstabilized filaments of TgAct1, we used phenix.douse to add waters outside of the magnesium-coordinated waters in the active site. While we observe waters in a hydration shell contacting each protomer, no waters are resolved in the center of the filament. The lack of waters may suggest a less organized water network within the filament.

Without the equivalent Mg²⁺-ADP-BeF³⁻ and Mg²⁺-ADP-Pi structures, we cannot perform the same analyses as Oosterheert et al. Comparison of unstabilized filaments of TgAct1 (Mg²⁺-ADP) to both Reynolds et al and Oosterheert et al filaments (Mg²⁺-ADP; PDB IDs 8d13 and 8a2t) shows TgAct1 filaments contain elements of both filaments' positioning of the nucleotide sensing residues. The Gln137 equivalent residue (Gln138) aligns best with Gln137 from 8a2t, but both residues Glu108 and Arg117 sit in between their equivalents (Glu107/Arg116) in 8d13 and 8a2t. The C-terminus of TgAct1 is most similar to 8d13, but shows an alternative position for the C-terminal Phe376. However, all the residues that form the sensing path from the nucleotide binding site to the C-terminus are conserved in TgAct1, so the sensing could act in the same way and be linked to the D-loop.

We now mention the possibility that both the water network and the nucleotide sensing residues may indirectly influence the D-loop in the results and discussion sections.

Line # 260: "The nucleotide sensing residue Gln138 (skeletal actin Gln137) and the residues that transmit nucleotide state to the periphery (Glu108 and Arg117; skeletal actin Glu107/Arg116) are

conserved in TgAct1. While the residues show positions that don't fully align with either chicken and rabbit MgADP actin filaments (PDB IDs 8d13 and 8a2t), they may still function to transmit nucleotide state information to the periphery (2022OosterheertNature) (Ext. Data Fig. 9c). [...] TgAct1 filaments do not contain an extensive water network within the center of the filament (Fig. Ext. Data Fig. 9e)."
Line # 346: "The structural basis for rapid disassembly is likely explained by the conformation and dynamics of the D-loop, although both the less ordered water network and the nucleotide sensing residues may indirectly influence the D-loop positioning and disassembly."

* Since interpretation of the map and model in the D-loop is important to the claims of this study, the model validation statistics need scrutiny. It seems there are some residues with poor fit in the vicinity of the D-loop for both structures, but especially the Jasp-bound model. The authors acknowledge this in the Results by stating a potential mixed occupancy for the Jasp-bound D-loop, but may oversell the D-loop change between the Jasp-unbound and Jasp-bound state as binary in the Discussion. For example, "Our structures reveal two distinct D-loop conformations for the stabilized vs. unstabilized structure" is slightly inaccurate, since there would be at least two D-loop conformations based on the data. Could the authors acknowledge this in the Discussion and balance this assertion?

We have updated the discussion to acknowledge the mixed occupancy.

Line # 348: "Our structures reveal at least two D-loop conformations for TgAct1; while we have modeled one conformation for each structure, the volumes suggest some mixed occupancy. In the jasplakinolide stabilized structure, the D-loop is primarily inserted into the binding pocket, whereas in the absence of stabilizing agents, the loop is partially out of the binding pocket."

* Furthermore on the Discussion regarding the D-loop: "It suggests that the loop makes the full set of contacts during filament assembly and shifts after assembly, decreasing the number of longitudinal interactions." The jasp-treated conformation might not necessarily reflect filament assembly in the untreated state. This sentence could be toned down to something more speculative.

We have updated the discussion to clarify that this statement is speculative.

Line # 352: "While speculative, one possibility is that the D-Loop could progress through different conformations as the filament ages, with the loop making the full set of contacts during filament assembly and then losing contacts during the aging process, decreasing the number of longitudinal interactions and speeding disassembly."

* The conclusions would be better supported by mutagenesis of the key residues discussed in the D-loop e.g. P42 and I44. Even more compelling would be a loop replacement with residues from mammalian skeletal actin or PfAct1 or PfAct2. But this is likely beyond the scope of this study.

While outside the scope of the current study, we agree that the mutagenesis experiments are compelling. In a study on Plasmodium actin 1, Lu and colleagues conducted a mutagenesis experiment using PfAct1 in their work published in PNAS in 2019 (<https://doi.org/10.1073/pnas.1906600116>). The authors substituted the PfAct1 D-loop with the canonical human actin D-loop and found that this substitution lowered the critical concentration for assembly of PfAct1. As the D-loop of PfAct1 and TgAct1 contain the same amino acids, we think that the results of substituting the canonical human actin D-loop for the D-loop of TgAct1 would produce similar results. We now specifically mention this published work in our comparison of TgAct1 to PfAct1.

Line # 332: "In contrast, biochemical work on *P. falciparum* Act1 suggests that the D-loop contributes to the instability of the filaments; substitution of the D-loop of *P. falciparum* Act1 with the canonical human actin D-loop decreased the critical concentration for assembly of *P. falciparum* Act1."

* Figure 4e would benefit from labels for the surrounding residues referred to in the Results text (e.g. residues

139-144, 169-173 and 352-370). The superimposed densities and models in Figure 4c are also quite unclear, perhaps it could be consolidated with 4e and enlarged. Readers can then better appreciate how this D-loop change is accommodated.

Please note that old Figure 4e is now 5e. We have updated figure panels 5c and 5e. In 5c, we now only show the comparison of the D-loop volumes and have labeled those residues mentioned in the text. In 5e, we have changed the representation into a ribbon and stick diagram, labeled the residues mentioned in the text, and added arrows that emphasize the changes we describe in the text.

Figure Legend: Figure 5 [...] c, Overlay of volume from the D-loops of TgAct1 + jasplakinolide (blue) and unstabilized TgAct1 (gray). [...] e, Overlay of the D-loop and binding pocket from unstabilized TgAct1 filaments (gray) and TgAct1 + jasplakinolide filaments (blue). Backbone shown as ribbon with select residues shown as sticks.

* Ext Data Fig 6e is quite unclear. Could it be enlarged and can H372 be labelled in the unstabilized TgAct1 model?

Please note that old Ext. Data Fig 6e is now Ext. Data Figure 7e. We have updated Ext. Data Figure 7 to enlarge the C-terminal comparison (now panel 7g), added labels for residues H372 and E361, and converted the backbone to a ribbon diagram for better clarity.

Figure Legend: Ext. Data Figure 7: [...] g, Volumes and models of the C-termini (unstabilized TgAct, top; TgAct1 + jasplakinolide, bottom).

Reviewer #2 (Remarks to the Author):

Actin is an incredibly well conserved polymer that appears to have originated in bacteria prior to specializing into a core component of the eukaryotic cytoskeleton. Some organisms encode divergent actin proteins with unusual biochemical properties. Apicomplexan parasites are an example of this, as they have a divergent actin that has been argued to form incredibly unstable filaments which cannot grow long. In this study, a collaboration between 3 research groups with combined expertise in actin biochemistry, cell biology, and structure, the authors carefully compare the biochemistry of Toxoplasma actin to the well-studied vertebrate skeletal actin. In contrast with previous reports, the authors find that TgACT1 is able to form long filaments, though it has the unusual property of treadmilling. The authors quantify the total actin levels in a Toxoplasma cell, which they find to be consistent with the in vitro critical concentration for TgACT1 polymer formation. Finally, they provide two high resolution cryo-EM structures of native, untagged TgACT1 with ATP and with the stabilizing agent jasplakinolide.

The paper is well-written and largely easy to follow. All experiments appear to have been carefully and rigorously carried out. While the paper is relatively short and the concepts discussed reasonably straightforward, I feel overall it is of high quality and adds to our understanding both of how apparent minor changes in actin can alter its biochemical properties and helps reconcile previously conflicting reports on Apicomplexan actin when comparing in vitro versus cellular data.

Overall, my comments are relatively minor.

-While cryo-EM structures are distinct from the previously published Plasmodium actin (Pospich 2017) in that the actin used is untagged and one structure has only non-analog ATP without any stabilizing agents, the structures of jas-stabilized could and should be directly compared.

We have added direct comparisons of Jasplakinolide-stabilized TgAct1 to Jasplakinolide-stabilized Plasmodium Act1 in main text Figure 5 and the Results section. For our comparison, we used PDB ID 6tu4 (2022 Vahokoski et al, PLoS Pathogens), as the structure was determined to significantly

higher resolution as compared to the first structure (PDB ID 5ogw from 2017 Pospich et al, Proc Natl Acad Sci USA): 2.6 Å for 6tu4 versus 3.8 Å for 5ogw.

Line # 310: “The TgAct1 protomers from jasplakinolide-bound filaments show high structural similarity to protomers from jasplakinolide-bound, *P. falciparum* Act1 filaments, with a C α RMSD value of 0.6 (residues 6-372, PDB ID 6tu4) (2017PospichPNAS; 2022VahokoskiPlosPath). The helical symmetries of filaments from TgAct1 and *P. falciparum* Act1 are very similar (*P. falciparum* Act1: 28.4 Å rise and -167.6° twist) (2022VahokoskiPlosPath). The D-loop of the jasplakinolide-bound filament adopts a conformation nearly identical to that of jasplakinolide-bound *P. falciparum* Act1 and skeletal actin loops (Fig. 5d and f). C α RMSD values comparing the D-loops demonstrate the overlap: 0.8 and 1.2 comparing jasplakinolide-bound TgAct1 to jasplakinolide-bound *P. falciparum* Act1 and native chicken skeletal actin, respectively.

Comparison of jasplakinolide-bound TgAct1 and *P. falciparum* Act1 reveals a few differences. Alignment of the protomers shows small changes in the positioning of the C-terminal backbone when comparing TgAct1 and *P. falciparum* Act1 (Fig. 5g). The unresolved C-terminus and the shift in the backbone position change the hydrogen bond network that C-terminal residues Ser369 and His372 make with residues Lys114, Glu117, and Arg118. As mentioned above, the D-loops contain the same residues; however, there are three amino acid substitutions immediately outside of the D-loop. In addition to Cys53 substitution mentioned above, TgAct1 contains Lys38, and Tyr54, while *P. falciparum* Act1 contains Arg38 and Phe54 (Ext Data Fig. 8). Overall, these are conservative substitutions; Lys38 and Tyr54 adopt similar positions to Arg38 and Phe54.”

-Some Figure panels seem out of order as described in the legend which made it a bit confusing following things (e.g. Figure 1).

The order in the legend was fixed.

While the methods section is overall appropriately detailed, the program used for curve fitting and statistical analysis was a bit vague; presumably Graphpad Prism, though this is not stated for fitting (“e.g. fitted to an exponential [in..]”) nor are any statistics provided, e.g. errors of fit, or statistics validating comparisons of curves or values displayed in bar graphs.

We added additional descriptions of error to Figure legend 2c.

Figure Legend: Figure 2: [...] c, Time course of phosphate release from 12 μ M TgAct1 (blue) and skeletal actin (orange) after inducing polymerization. Error bars represent standard deviation of the mean for three independent experiments.

We also added further details on our statistical analysis and equations of fit to the methods section.

Line # 552: “Time courses of fluorescence change were fitted to single exponential decay in the form of $F_{\text{obs}}(t) = \text{Amp}(1 - e^{-kt}) + F_{\text{IO}}(t_0)$, where $F_{\text{obs}}(t)$ is the observed fluorescence intensity signal at time t , Amp is the amplitude of the fluorescence signal change (negative value for an exponential decay), $F_{\text{IO}}(t_0)$ is the initial fluorescence intensity signal at time $t = 0$, and k is the observed ϵ ATP dissociation rate constant (66), using GraphPad Prism (GraphPad Software ver. 10.0.0 (Boston, Massachusetts USA). Because nucleotide dissociation is rate-limiting and nucleotide binding is far more rapid the observed rate constant of the best fit yields the nucleotide dissociation rate constant. Uncertainties associated with the average nucleotide dissociation rate constants are represented as the standard deviation of the mean for three independent experiments.”

Two issues with “This isoform shares only 83% similarity to skeletal alpha actin and mammalian β and γ isoforms (4)”. 1) “percent identity” is not meaningful without noting the substitution matrix used to compare. Normally, the simplest and most robust comparison is percent identity. 2) is it correct to call TgAct1 an “isoform” if there’s only one actin-encoding gene in that organism and you are comparing to different organisms? Ortholog or simply “protein” seems more appropriate here.

83% is the percent identity. Similarity was a typo and we have updated the text. Also, isoform has been changed to protein.

It seems appropriate to include the original reference (Mondragon & Frixione 1996) that implicated actin in conoid movement in addition to the more recent paper referenced.

We have added the reference.

Figure 1e – I see no blue dashed line in this figure.

We have decreased the opacity of the blue line so that it is visible.

Reviewer #3 (Remarks to the Author):

In this work, Hvorecny and Sladewski et al. analyzed the kinetics and structure of Toxoplasma actin, which only shares 83% similarity to the skeletal alpha actin, the canonical model for actin biology. This work revealed the extraordinarily high critical concentration of TgAct1, which can be explained by reduced contact sites with the protein backbone around the D-loop. The nucleotide exchange rate of TgAct1 was found to be ~ 50 fold higher than that of the skeletal actin. This can be explained by a single substitution within the nucleotide-binding pocket between the skeletal muscle actin and TgAct1, from a leucine (Leu16) to an asparagine (Asn17). The manuscript is very well written and the results are clearly presented. It for the first time provides a coherent view of how the basic biochemical properties of TgAct1 is related to its function in vivo. By carefully delineating the optimization of various experimental conditions, this work also effectively clarifies a number of long-standing confusions in the field regarding the polymerization kinetics of TgAct1. I have only a few comments for the authors to address.

1. The effect of Actin Chromobody (Cb) on actin polymerization

The effect of Actin Chromobody (Cb) on polymerization of TgAct1 is somewhat a debatable point. Here the authors showed that the measurements of Cc, disassembly and assembly rate constants are consistent with previously reported values for the skeletal actin. However the effect of Cb on TgAct1 polymerization remains untested. Since Cb is the most commonly used probe for labeling actin in Toxoplasma, it would be tremendously helpful for the field if the authors could directly compare the kinetics of TgAct1 polymerization with or without Cb using either light scattering or cryoEM analysis.

We have observed that apicomplexan actin forms small oligomers over time when not stabilized with ammonium acetate. These oligomers can scatter light which can complicate the interpretation of light scattering results, making it difficult to see differences in bulk polymerization kinetics. Because of this, we used a microscopy-based endpoint assay to study the effect of the chromobody on TgAct1 polymerization kinetics. Briefly, TgAct1 was induced for polymerization in the presence or absence of 50 μ M chromobody and allowed to polymerize in flow chambers. After 3, 4, or 7 minutes of polymerization, the assembly of actin was stopped by washing out TgAct1 monomers. Then, a solution containing the chromobody was added to image the resulting filaments. Without monomers present, further assembly is impossible, allowing us to visualize actin that was polymerized in the absence of the chromobody.

Actin lengths were measured at three timepoints for three different concentrations of TgAct1. The endpoint measurement generates a distribution of lengths at each timepoint, because actin polymerization is a stochastic process. Long filaments are those that started polymerizing right after induction, while short filaments initiated toward the end of the incubation. We find that the actin length distributions at all timepoints and at all concentrations are similar +/- chromobody. This indicates to us that the chromobody has no effect on the polymerization rate TgAct1 over the

timepoints and concentrations used. In another words, if the chromobody enhanced the polymerization kinetics of TgAct1 we would expect to see longer filaments in samples polymerized in the presence of chromobody, but we see no indication of this. To quantify this result, we determined average rates with or without chromobody by dividing the maximum actin filament lengths (corresponding to filaments that started polymerizing at the start of induction) by the incubation timepoint and converting this value to subunits per sec. We averaged the three timepoints together for each actin concentration. From this we found no significant difference in the rate of actin growth with or without chromobody. Actin length distributions, an example montage, and average polymerization rate analysis is presented in Ext. Data Fig. 3 and Ext. Data Table 1. This figure is addressed in the main text following the skeletal actin chromobody control in the results section.

Line # 136: "To verify that the chromobody has no effects on the rates of TgAct1 assembly, we measured length distributions for TgAct1 assembled in the presence or absence of actin chromobody at different time intervals for three different concentrations of actin using an endpoint actin assembly assay (Ext. Data Fig. 2). From this analysis, we find no detectable differences in length distributions with or without chromobody for all time intervals and actin concentrations tested, consistent with the chromobody having no effect on the rate of actin growth (Ext. Data Fig. 3a-b, Ext. Data Table 1). To quantify these results, we determined an average actin assembly growth rate for each actin concentration with or without chromobody. There was no significant difference in assembly rates due to the chromobody at all actin concentrations tested (Ext. Data Fig. 3c).

2. As this work revealed, much of the prior confusions in the field arose from suboptimal polymerization conditions. I therefore think that Ext Figure 3 should be included as a figure in the main text, as it shows beautifully how the key parameters that need to be tuned for reliable polymerization of TgAct1 can be determined.

We have moved panels e and f from Ext Data Figure 3 to main text Figure 3 (now panels a and b). As panels a and b from the original Ext. Data Figure 3 repeat experiments published in the literature, panel c uses AMPPNP, and panel d shows negative data, we chose to keep these panels as Ext Data Figure 5.

3. The legends for panel f and g in Figure 1 are reversed.

The legend is fixed.

4. Page 7, 4th paragraph: "shift after assembly" should be "shift after disassembly"

To address a comment from Reviewer 1, this sentence has been rewritten.

Line # 352: "While speculative, one possibility is that the D-Loop could progress through different conformations as the filament ages, with the loop making the full set of contacts during filament assembly and then losing contacts during the aging process, decreasing the number of longitudinal interactions and speeding disassembly."

5. Page 7, 4th paragraph: "The D-loop differences between stabilized and unstabilized TgAct1 filaments helps to reconcile why previous structures of P. falciparum Act1 could not fully explain the importance of the D-loop in filament stability."

I don't follow this argument. It seems to me that the importance of the D-loop in filament stability for P. falciparum Act1 is still unexplained.

Agreed. We pointed out that no differences in the position of the D-loop have been observed in

previous structures of Plasmodium Act1, but do not further speculate as to the importance of D-loop in Plasmodium actin or why differences in its position have not been observed.

Line # 361: “However, despite the known importance of the D-loop in regulating P. falciparum Act1 filament stability, previous structures of P. falciparum Act1 stabilized with jasplakinolide observed minimal differences in the position of the D-loop of P. falciparum Act1 relative to skeletal muscle actin (51).”

6. Line numbers should be included for the manuscript text.

We have added line numbers to the manuscript text.

REVIEWERS' COMMENTS

Reviewer #1 (Remarks to the Author):

My comments have been addressed, thank you!

Reviewer #3 (Remarks to the Author):

The authors have addressed all my concerns.